# Dissociation rate compensation mechanism for budding yeast pioneer transcription factors

Benjamin T Donovan[1], Hengye Chen[2,3], Caroline Jipa[4], Lu Bai[2,5], Michael G Poirier[1,4,6,7]*

[1]Biophysics Graduate Program, The Ohio State University, Columbus, United States; [2]Department of Biochemistry and Molecular Biology, The Pennsylvania State University, State College, United States; [3]Center for Eukaryotic Gene Regulation, The Pennsylvania State University, State College, United States; [4]Department of Physics, The Ohio State University, Columbus, United States; [5]Department of Physics, The Pennsylvania State University, State College, United States; [6]Ohio State Biochemistry Graduate Program, The Ohio State University, Columbus, United States; [7]Department of Chemistry and Biochemistry, The Ohio State University, Columbus, United States

**Abstract** Nucleosomes restrict the occupancy of most transcription factors (TF) by reducing binding and accelerating dissociation, while a small group of TFs have high affinities to nucleosome-embedded sites and facilitate nucleosome displacement. To understand this process mechanistically, we investigated two *Saccharomyces cerevisiae* TFs, Reb1 and Cbf1. We show that these factors bind to their sites within nucleosomes with similar binding affinities as to naked DNA, trapping a partially unwrapped nucleosome without histone eviction. Both the binding and dissociation rates of Reb1 and Cbf1 are significantly slower at the nucleosomal sites relative to those for naked DNA, demonstrating that the high affinities are achieved by increasing the dwell time on nucleosomes in order to compensate for reduced binding. Reb1 also shows slow migration rate in the yeast nuclei. These properties are similar to those of human pioneer factors (PFs), suggesting that the mechanism of nucleosome targeting is conserved from yeast to humans.
DOI: https://doi.org/10.7554/eLife.43008.001

*For correspondence:
poirier.18@osu.edu

**Competing interests:** The authors declare that no competing interests exist.

## Introduction

The fundamental unit of chromatin is the nucleosome, ~147 base pairs (bp) of DNA wrapped around a core of eight histone proteins (*Luger et al., 1997*). Extensive contacts between nucleosomal DNA and the histone octamer suppress access to DNA-binding proteins, including many transcription factors (TFs) (*Wolffe, 1992*). To overcome this steric occlusion, TFs take advantage of dynamic nucleosome structural fluctuations, which transiently expose DNA-binding sites for TF binding. However, this site exposure mechanism (*Li and Widom, 2004*; *Polach and Widom, 1995*) results in reduced occupancy relative to that on naked DNA, since the TF can bind only when the site is partially unwrapped. In addition, it was recently shown that nucleosomes increase TF dissociation rates by orders of magnitude (*Luo et al., 2014b*). In combination, the decreased binding and increased dissociation rates can result in a reduction in the apparent dissociation constant for nucleosome substrates of over a 1000-fold. For example, Gal4 binds to its DNA target site at picomolar concentrations (*Liang et al., 1996*) whereas it requires nanomolar concentrations to bind nucleosomal DNA (*Luo et al., 2014b*). In contrast to TFs such as Gal4, pioneer transcription factors (PFs) access their

binding sites within nucleosomes as efficiently as they access fully exposed DNA without the aid of additional factors (*Zaret and Carroll, 2011*). This property is thought to allow PFs to target closed chromatin and to prime transcription activation (*Iwafuchi-Doi and Zaret, 2014*).

In budding yeast, chromatin is mainly opened by a few highly expressed TFs that can access their nucleosome-embedded binding sites in the genome and establish local nucleosome depleted regions (NDRs) (*Yan et al., 2018*). How these TFs gain access to their DNA target sites and facilitate nucleosome displacement is not well understood. Two non-exclusive mechanisms may be used during this process. With a ''passive'' mechanism, TFs can occupy naked DNA when the nucleosome structure is temporarily disrupted by another cellular event (e.g. DNA replication). Note that this mechanism does not require TFs to interact with nucleosomes. Alternatively, TFs may directly bind and invade into nucleosomes. The key to differentiating between these two models is to determine whether certain TFs, such as PFs, can stably engage a nucleosomal template containing their recognition sites (*Hughes and Rando, 2014*).

One well-studied nucleosome-depleting TF is Reb1, a factor essential for yeast viability. Consistent with its ability to displace nucleosomes, most of the Reb1-binding sites in the genome reside in NDRs (*Lee et al., 2007*). However, ~20% (154/903) of Reb1-binding sites exist within well-positioned nucleosomes and almost half (71/154) of these sites are occupied by Reb1. Reb1 tends to bind the nucleosome near the entry-exit site and has been shown to increase the accessibility of DNA locally (*Koerber et al., 2009*). Overall, these observations suggest that Reb1 may gain access to nucleosomes near the entry-exit site through the site exposure model (*Koerber et al., 2009*; *Polach and Widom, 1995*), but the stability and the kinetics of this interaction are unknown.

In this study, we used a combination of in vitro techniques, including gel electromobility shift assays (EMSA), ensemble fluorescence, and single-molecule fluorescence, to determine whether and how Reb1 invades nucleosomes. We found that Reb1 accesses its site in both DNA and nucleosomes with similar affinities and targets entry-exit sites by trapping the nucleosome in a partially unwrapped state without evicting histones. As for other TFs, nucleosome site exposure lowers the Reb1 association rate when binding to nucleosome-embedded sites. However, once bound, Reb1 compensates for the reduced association rate with an equally reduced dissociation rate. These properties may be general among nucleosome-displacing factors: we show that another *Saccharomyces cerevisiae* TF, Cbf1, binds nucleosome with similar dynamics. Finally, in vivo fluorescence recovery after photobleaching (FRAP) experiments indicate that Reb1 undergoes exchange within the nucleus that is markedly slower than that of other chromatin-interacting proteins. These properties were previously reported for the human PFs (*Cirillo and Zaret, 1999*; *Sekiya et al., 2009*). We therefore propose that Reb1 and Cbf1 can function as PFs by using a dissociation rate compensation mechanism to target nucleosomes efficiently, to partially unwrap nucleosomes, and to facilitate the recruitment of transcription regulatory complexes to define NDRs and to activate transcription.

## Results

### Reb1 binds DNA and nucleosomes with similar affinities

A defining property of PFs is that nucleosomes do not impede their binding. To determine whether Reb1 exhibits this characteristic, we quantified Reb1 affinities to both nucleosome and DNA substrates. For DNA binding experiments, we tested binding to 25-bp oligos containing the Reb1-binding motif. With reconstituted, sucrose gradient purified nucleosomes (*Figure 1—figure supplement 1*), we tested binding to entry-exit sites because this is where Reb1 preferentially binds in vivo (*Koerber et al., 2009*). We tested binding at four separate sites positioned in increments of 5 bp throughout the entry-exit region (*Figure 1A*). We refer to these templates as 'Px,' where 'x' defines the beginning of the Reb1-binding site in the 601 nucleosome positioning sequence (NPS) (i.e. P3 = binding site starts 3 bp into the nucleosome). Binding to both DNA and nucleosomes was detected via EMSA, in which we titrate Reb1 and observe formation of a slow-mobility Reb1 complex (*Figure 1B–C*). For Reb1 binding to nucleosomes, we imaged EMSAs with Cy5-H2A(K119C) and Cy3-DNA fluorescence (*Figure 1C*, *Figure 1—figure supplement 2*), confirming that Reb1 is in complex with nucleosomes. Affinity was measured for each binding reaction by determining the $S_{1/2}$, the concentration at which 50% of the DNA or nucleosomes are bound by Reb1. For DNA, we measured $S_{1/2 \text{ Reb1–DNA + site EMSA}} = 2.3 \pm 0.2$ nM, whereas for the four nucleosome constructs, we

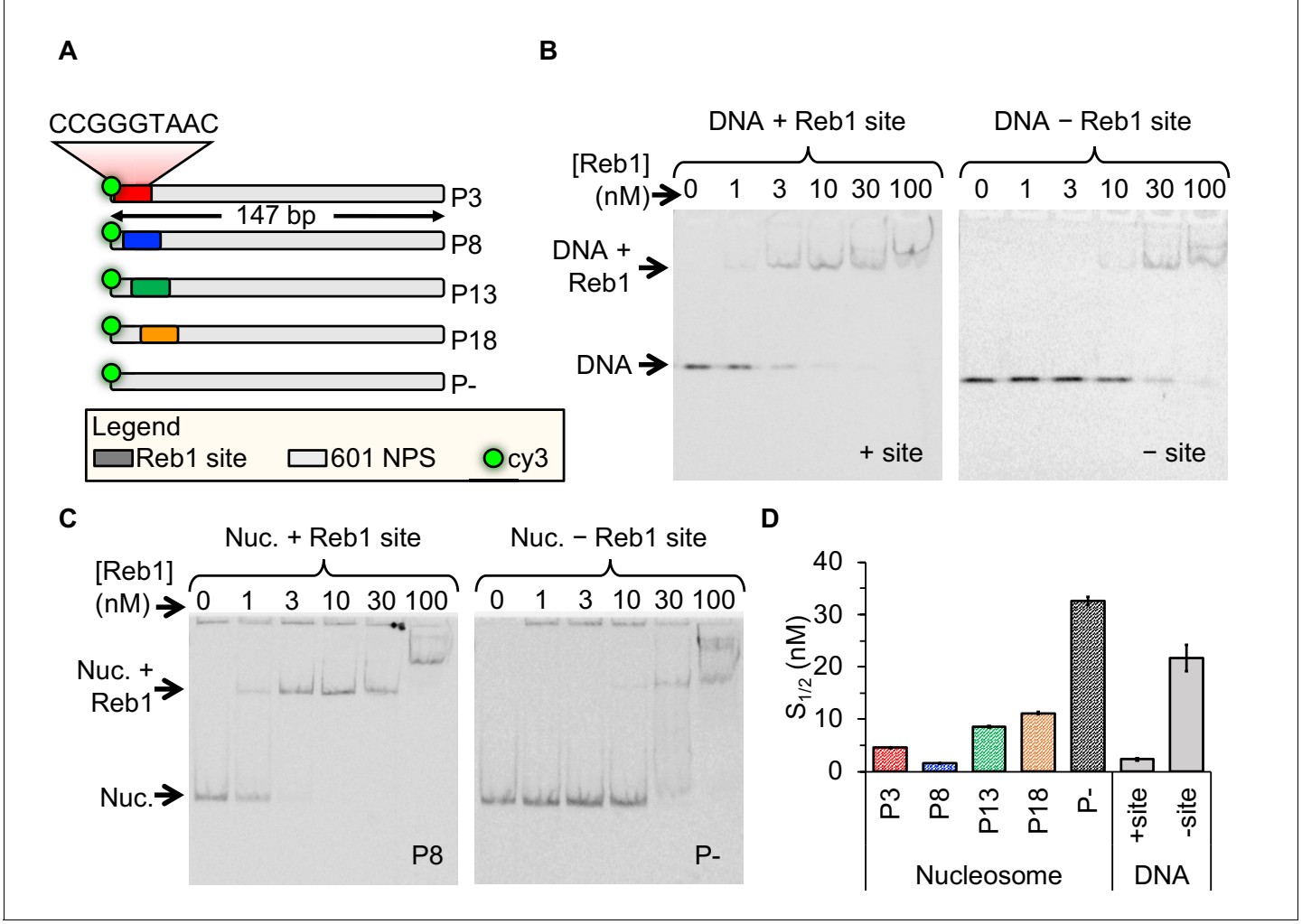

**Figure 1.** Reb1 binds DNA and nucleosomes with similar affinities. (**A**) Design of the modified '601' nucleosome positioning sequences (NPS) used in this study. Colored rectangles represent the Reb1-binding site at positions P3 (red), P8 (blue), P13 (green) and P18 (gold) within the 601 NPS. The numbers indicate the starting position of the Reb1-binding site (in number of base pairs into the 601 NPS). (**B**) Cy3 image of the EMSA of Reb1 binding to a 25-bp DNA sequence with (left) or without (right) the Reb1-binding site. (**C**) Cy5 image of the EMSA of Reb1 binding to P8 nucleosomes with (left) or without (right) the Reb1-binding site. (**D**) Quantification of the $S_{1/2}$s determined from the Reb1 EMSAs in panels (**B**) and (**C**) and in *Figure 1—figure supplement 3* ($S_{1/2\ Reb1–DNA\ +\ site\ EMSA}$ = 2.3 ± 0.2 nM, $S_{1/2\ Reb1–DNA\ –\ site\ EMSA}$ = 21.7 ± 2.3 nM, $S_{1/2\ Reb1–Nuc\ P3\ EMSA}$ = 4.6 ± 0.1 nM, $S_{1/2\ Reb1–Nuc\ P8\ EMSA}$ = 1.5 ± 0.1 nM, $S_{1/2\ Reb1–Nuc\ P13\ EMSA}$ = 8.5 ± 0.2 nM, $S_{1/2\ Reb1–Nuc\ P18\ EMSA}$ = 11.2 ± 0.3 nM, $S_{1/2\ Reb1–Nuc\ P\ –\ EMSA}$ = 32.7 ± 0.8 nM). These results show that Reb1 binds nucleosomes and DNA sites specifically with a similar $S_{1/2}$.

DOI: https://doi.org/10.7554/eLife.43008.002

The following figure supplements are available for figure 1:

**Figure supplement 1.** Nucleosomes and TFs used in this study.
DOI: https://doi.org/10.7554/eLife.43008.003

**Figure supplement 2.** The Reb1–nucleosome bound complex in EMSAs.
DOI: https://doi.org/10.7554/eLife.43008.004

**Figure supplement 3.** Reb1 nucleosome EMSAs.
DOI: https://doi.org/10.7554/eLife.43008.005

**Figure supplement 4.** Reb1–nucleosome EMSA fits.
DOI: https://doi.org/10.7554/eLife.43008.006

**Figure supplement 5.** Reb1–DNA EMSAs.
DOI: https://doi.org/10.7554/eLife.43008.007

**Figure supplement 6.** Comparison of Reb1-binding affinities to labeled and unlabeled octamers.
DOI: https://doi.org/10.7554/eLife.43008.008

measured $S_{1/2\ Reb1-Nuc\ P3\ EMSA}$ = 4.6 ± 0.1 nM, $S_{1/2\ Reb1-Nuc\ P8\ EMSA}$ = 1.5 ± 0.1 nM, $S_{1/2\ Reb1-Nuc\ P13\ EMSA}$ = 8.5 ± 0.2 nM, and $S_{1/2\ Reb1-Nuc\ P18\ EMSA}$ = 11.2 ± 0.3 nM. In addition, we performed EMSAs with DNA and nucleosomes lacking the specific binding site and measured ~10-fold lower affinity for these sequences [$S_{1/2\ Reb1-DNA\ -site\ EMSA}$ = 21.7 ± 2.3 nM and $S_{1/2\ Reb1-Nuc\ P-EMSA}$ = 32.7 ± 0.8 nM (*Figure 1D*, *Figure 1—figure supplement 3*, *Figure 1—figure supplement 4*, *Figure 1—figure supplement 5*, *Supplementary file 1* Table S1). Reb1's targeting of DNA and nucleosome with similar affinities mimics the behavior of PFs in higher eukaryotes (*Soufi et al., 2015*). By contrast, other TFs that employ the site exposure model to invade nucleosome entry-exit sites, such as Gal4 and LexA, require over 1,000-fold higher TF concentrations to bind these sites relative to binding naked DNA (*Luo et al., 2014b*).

## Reb1 invades nucleosomes by trapping entry-exits sites in a partially unwrapped state

Previous genome-wide studies of nucleosome and Reb1 occupancy suggest that Reb1 gains access to nucleosome entry-exit sites via the mechanism described by the site exposure model (*Koerber et al., 2009*). We investigated this binding mechanism through a series of fluorescence resonance energy transfer (FRET) experiments that monitored nucleosomes trapped by Reb1 in partially unwrapped states (as previously done for other TFs [*Gibson et al., 2016*; *Li et al., 2005*; *Li and Widom, 2004*]). The donor fluorophore Cy3 was attached to the 5′-end of the DNA adjacent to the Reb1-binding site while the acceptor fluorophore Cy5 was attached to H2A(K119C) (*Figure 2A*). The proximity of these two locations within the nucleosome results in high FRET efficiency (85%; *Figure 2B*). For P3 and P8 nucleosomes, titrating Reb1 progressively lowers the FRET efficiency, with saturation occurring at ~20%. This relationship fits to a binding isotherm with $S_{1/2}$ values ($S_{1/2\ Reb1-Nuc\ P3\ FRET}$ = 7.9 ± 1.3 nM, $S_{1/2\ Reb1-Nuc\ P8\ FRET}$ = 2.4 ± 0.3 nM; *Figure 2C*, *Supplementary file 1* Table S1) that agree with $S_{1/2}$ values for the corresponding EMSA measurements (*Figure 1C*). The similarity of the $S_{1/2}$ values for Reb1 binding and ΔFRET strongly suggests that Reb1 binding to its target site causes a significant structural change in the nucleosome. By contrast, significantly higher concentrations of Reb1 are required to induce a ΔFRET with P13 nucleosomes ($S_{1/2\ Reb1-Nuc\ P13\ FRET}$ = 101.5 ± 19.1 nM), in fact these values were ~12 fold higher than the concentration measured by EMSA for Reb1–nucleosome binding (*Figure 1D*). In addition, we did not observe a significant ΔFRET for P18 nucleosomes. This indicates that Reb1 can bind to sites further into the nucleosomes but does not induce a structural change. Finally, we demonstrated that our observed ΔFRET is site specific, as Reb1 titrations with nucleosomes that do not contain a binding site result in no ΔFRET (*Figure 2B*).

Reb1 is reported to occupy nucleosomes at the entry-exit region in vivo and to reduce the nucleosome footprint by 12 base pairs (*Koerber et al., 2009*). Therefore, we focused on the two Reb1 positions closest to where the nucleosomal DNA enters/exits the nucleosome, the P3 and P8 nucleosomes, where the Reb1 site extends 12 and 17 base pairs into the nucleosome, respectively. We carried out additional experiments to determine the nature of the Reb1-induced ΔFRET. First, by separately imaging with fluorescence from Cy3-DNA and Cy5-H2A(K119C) in the EMSAs of Reb1-nucleosome binding, we demonstrated that Reb1 is in complex with nucleosomes. Therefore, the ΔFRET is not dependent on partial or full nucleosome disassembly (*Figure 1A–B*, *Figure 1—figure supplements 2* and *3*).

It is possible that the observed ΔFRET is due to Reb1-induced structural changes in the H2A C-terminal tail that occur upon binding, as the Cy5 fluorophore is positioned in this domain (*Figure 2—figure supplement 1A*). To test for this, we prepared nucleosomes with a Cy5 fluorophore positioned at H3(V35C). Titrating Reb1 reveals similar nucleosome binding and ΔFRET (*Figure 2—figure supplement 1B–C*), which rules out the possibility that the ΔFRET is due to a distortion in the H2A C-terminal domain and suggests that the Reb1-induced structural change involves the DNA and the entire octamer.

Another potential explanation for the Reb1-induced structural change is that it traps repositioned nucleosomes. To test this idea, we inserted the Reb1-binding site onto the opposite side of the nucleosome from the Cy3 fluorophore and a 20-bp flanking sequence (*Figure 2—figure supplement 2A*). If Reb1-induced nucleosome structural change is due to octamer translocation, we would have detected a decrease in FRET that corresponds with Reb1 binding and with repositioning the octamer onto the flanking sequence. Although we detected Reb1 binding with EMSA, no ΔFRET

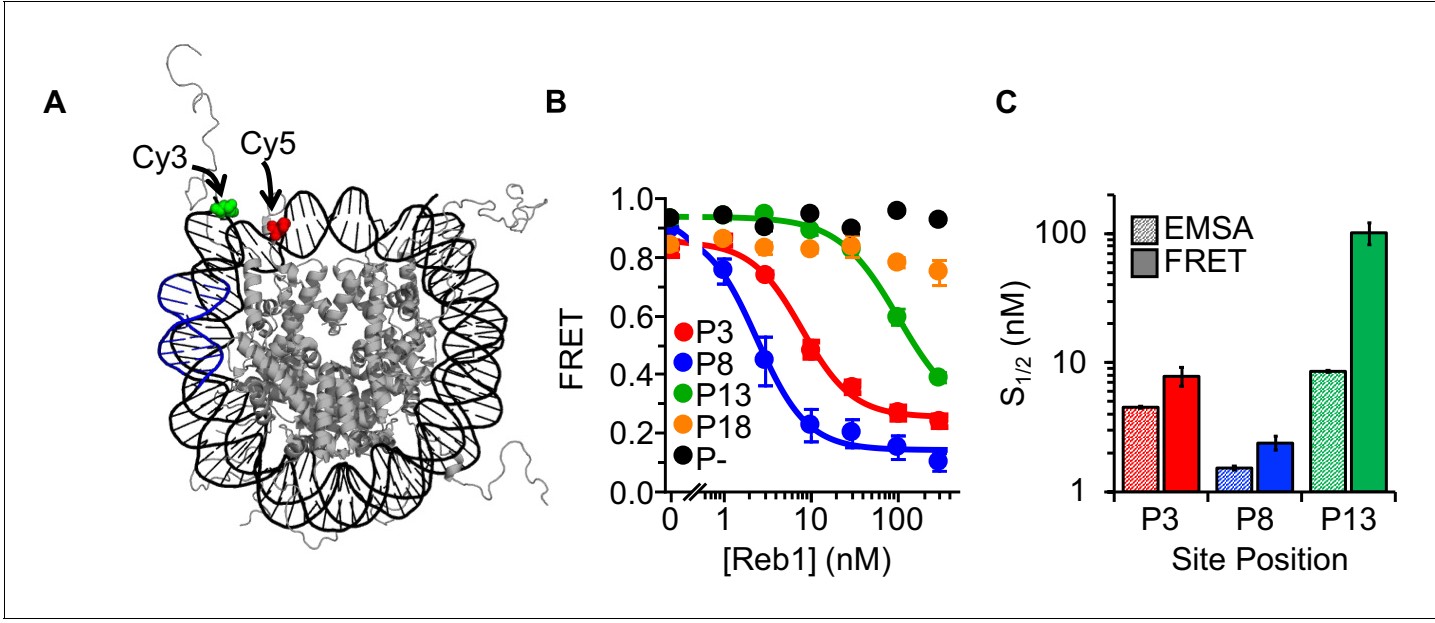

**Figure 2.** Reb1 binding induces nucleosome structural change. (**A**) Nucleosome structure (*Davey et al., 2002*) containing the internal FRET pair used in this study. Cy3 is attached to the 5′ end of the DNA NPS and adjacent to the Reb1 binding site (blue). The octamer is labeled with Cy5 at H2A(K119C). When fully wrapped, the nucleosome is in a high FRET state. (**B**) Nucleosome FRET efficiency measurements while titrating Reb1 with the nucleosome constructs: P3 (red), P8 (blue), P13 (green), P18 (gold), or no binding site (P–, black). Reb1 titrations with P3, P8, and P13 nucleosomes fit to binding isotherms with $S_{1/2 \text{ Reb1–Nuc P3 FRET}} = 7.9 \pm 1.3$ nM, $S_{1/2 \text{ Reb1–Nuc P8 FRET}} = 2.4 \pm 0.3$ nM, $S_{1/2 \text{ Reb1–Nuc P13 FRET}} = 101.5 \pm 19.1$ nM. We do not observe a significant ΔFRET for P18 and P– nucleosomes. (**C**) Comparison of the $S_{1/2}$ values obtained from EMSA and FRET experiments. For P3 and P8 nucleosomes, the FRET $S_{1/2}$ values are in close agreement to the EMSA $S_{1/2}$ values, indicating that ΔFRET is a measure of Reb1 binding to nucleosomes.

DOI: https://doi.org/10.7554/eLife.43008.009

The following figure supplements are available for figure 2:

**Figure supplement 1.** Reb1-induced nucleosome ΔFRET is not the result of structural changes in the H2A C-terminal domain.
DOI: https://doi.org/10.7554/eLife.43008.010

**Figure supplement 2.** Reb1 does not trap nucleosomes in a repositioned state.
DOI: https://doi.org/10.7554/eLife.43008.011

was observed (*Figure 2—figure supplement 2B–C*), suggesting that Reb1 binding does not result in repositioned nucleosomes.

These combined results support the conclusion that Reb1 binds to its site within the nucleosome entry-exit region via the mechanism described by the site exposure model, in which Reb1 traps the nucleosome in a partially unwrapped state. Interestingly, a direct conclusion from the site exposure model is that the Reb1-binding rate should be reduced as the probability that the site is exposed decreases, which in this region of the nucleosome is about 100-fold (*Li et al., 2005*). Therefore, the site-exposure model alone cannot explain why partially unwrapped nucleosomes and naked DNA are equally accessible to Reb1.

## Reb1 rapidly binds and dissociates at fully exposed DNA-binding sites

To investigate how Reb1 can trap a nucleosome in a partially unwrapped state and bind to it with the same affinity as it binds DNA, we used single molecule total internal reflection fluorescence (smTIRF) microscopy to measure the binding and dissociation kinetics of Reb1's interactions with its binding sites within both DNA and nucleosomes. Reb1–DNA binding was probed using protein induced fluorescence enhancement (PIFE), as performed previously for other TFs (*Gibson et al., 2016*; *Hwang et al., 2011*; *Luo et al., 2014b*). Here, the Reb1-binding site was positioned 1 bp away from a Cy3 fluorophore on the 5′ end of the DNA (*Figure 3A*). Titrating Reb1 induces a 1.5-fold increase in Cy3 fluorescence emission, which fits to a binding isotherm with an $S_{1/2 \text{ Reb1–DNA PIFE}}$ of $5.1 \pm 0.2$ nM (*Figure 3B*), whereas without the binding site, Cy3 fluorescence does not increase

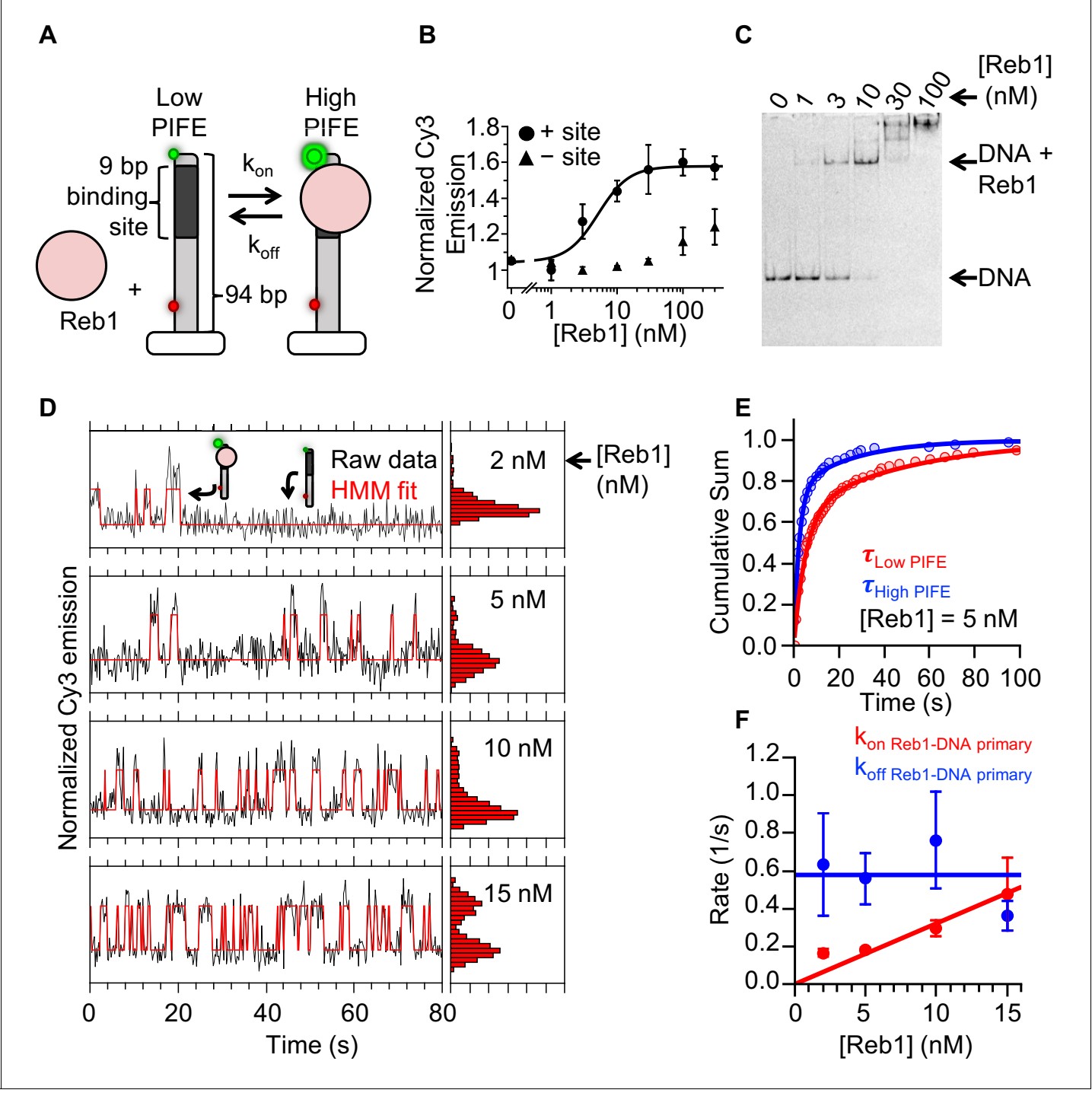

**Figure 3.** Reb1 rapidly binds to and dissociates from fully exposed DNA-binding sites. (**A**) Design of the smPIFE measurements. The 94-bp DNA molecule with the Reb1-binding site 1 bp from the Cy3-labeled 5′ end was immobilized on a quartz surface through a biotin–streptavidin linkage. DNA molecules are also Cy5-labeled, and we only analyzed molecules with signals in both Cy3 and Cy5. (**B**) Reb1 titration with the smPIFE DNA results in a Cy3 emission increase of ~1.5-fold and fits to a binding isotherm with an $S_{1/2\ \text{Reb1–DNA PIFE}}$ = 5.1 ± 0.2 nM. Without the binding site, the Cy3 emission does not change until 100 nM, demonstrating that the observed PIFE is due to site-specific Reb1 binding. (**C**) Cy3 image of the EMSA of Reb1 binding to the smPIFE DNA molecule. Reb1 binding is similar to that observed for the 25 bp DNA molecule ($S_{1/2\ \text{Reb1–DNA EMSA}}$ = 3.4 ± 0.1 nM). (**D**) Example time traces of single DNA molecules with four separate Reb1 concentrations, where the black lines are the Cy3 fluorescence and the red lines are the two-state Hidden Markov Model (HMM) fits. As the Reb1 concentration increases, the immobilized DNA molecules shift to the high PIFE state. (**E**) Example cumulative sums of low PIFE (red) and high PIFE (blue) dwell times that are fit with double exponentials. The Reb1 concentration is 5 nM. (**F**) The primary Reb1–DNA binding (red) and dissociation (blue) rates at four Reb1 concentrations. The dissociation rates are constant with an average rate

*Figure 3 continued on next page*

*Figure 3 continued*

of $k_{off\ Reb1–DNA\ primary}$ = 0.58 ± 0.08 s$^{-1}$, while the binding rate increases with Reb1 concentration. The overall binding rate is determined by fitting to a line whose slope represents the binding rate, $k_{on\ Reb1–DNA\ primary}$ = 0.032 ± 0.003 s$^{-1}$ nM$^{-1}$.

DOI: https://doi.org/10.7554/eLife.43008.012

The following figure supplement is available for figure 3:

**Figure supplement 1.** Reb1-binding and -dissociation rates for interactions with DNA.

DOI: https://doi.org/10.7554/eLife.43008.013

until 100 nM. This agrees with the EMSA binding $S_{1/2}$ (*Figure 3C*), demonstrating that PIFE detects site-specific Reb1 binding.

Next, we performed smTIRF measurements to determine Reb1-binding and -dissociation kinetics in the interaction with DNA (*Gibson et al., 2016*; *Roy et al., 2008*). The DNA was immobilized on a quartz microscope slide through a biotin–streptavidin linkage and included an 84-bp DNA extension to minimize surface interactions. In addition, to ensure that the Cy3 signal is due to a surface-tethered DNA molecule, we incorporated an internal Cy5-fluorophore adjacent to the biotin (*Figure 3A*). We only analyzed molecules with both a Cy3 and a Cy5 signal.

We measured the time-dependent fluorescence from at least 150 single molecules at four separate Reb1 concentrations: 2 nM, 5 nM, 10 nM, and 15 nM. These traces fluctuated between a high and a low Cy3 fluorescence emission state (*Figure 3D*), and the time spent in the high Cy3 emission state increased with Reb1 concentration. This indicates that the PIFE is sensitive to Reb1 binding and dissociation, as observed for other TFs (*Gibson et al., 2016*; *Luo et al., 2014b*).

To characterize the Reb1 binding and dissociation rates at each Reb1 concentration, we compiled the high and low PIFE dwell times into separate cumulative sums and fitted these cumulative sums to exponential distributions (*Figure 3—figure supplement 1A*). We used the low PIFE dwell time cumulative sum (*Figure 3E* and *Figure 3—figure supplement 1A*) to determine Reb1's binding rate to DNA. At each Reb1 concentration, the cumulative sum fits best to a double exponential (*Figure 3—figure supplement 1B*), where ~ 75% of unbound times are in the faster population (*Figure 3—figure supplement 1D*). This primary rate increases with Reb1 concentration (*Figure 3F*), where the slope of the linear fit gives an overall binding rate of $k_{on\ Reb1–DNA\ primary}$ = 0.032 ± 0.003 s$^{-1}$ nM$^{-1}$. By contrast, the secondary rate ($k_{on\ Reb1-DNA\ secondary}$ = 0.022 ± 0.002 s$^{-1}$) is not dependent on Reb1 concentration (*Figure 3—figure supplement 1C*), suggesting that it does not represent Reb1 binding. Instead, this secondary rate is probably due to a structural change in the Reb1–DNA complex that results in a transition from a low to a high PIFE state. The interpretation of these two types of low to high PIFE transitions is similar to those in previous studies of intrinsically disordered proteins, where one ON rate is concentration-dependent and interpreted as binding, while the second ON rate is concentration-independent and interpreted as a structural transition (*Dogan et al., 2012*). Reb1 may go through such transitions because of its large disordered domain (*Rost et al., 2004*).

We next analyzed the high PIFE dwell time cumulative sums to determine the Reb1 dissociation rate from DNA (*Figure 3E*). Log-likelihood ratio tests indicated that the dwell time histograms fit significantly better with double exponential distributions (*Figure 3—figure supplement 1B*). For all Reb1 concentrations, both rates do not depend on Reb1 concentration (*Figure 3F*, *Figure 3—figure supplement 1C*). ~75% of dwell times are associated with the faster rate of $k_{off\ Reb1–DNA\ primary}$ = 0.58 ± 0.08 s$^{-1}$ with a $\tau_{bound}$ = $1/k_{off}$ ≈ 1.7 s (*Figure 3—figure supplement 1D*, *Supplementary file 1* Table S2), whereas the remaining 25% are associated with a slower rate of $k_{off\ Reb–DNA\ secondary}$ = 0.036 ± 0.005 s$^{-1}$ with a $\tau_{bound}$ ≈ 28 s.

We compared the ratios of the primary dissociation rate (0.58 s$^{-1}$) or the secondary dissociation rate (0.036 s$^{-1}$) to the binding rate (0.032 s$^{-1}$ nM$^{-1}$) with the ensemble $S_{1/2}$ measurement, since these ratios are the apparent dissociation constants, $K_D$. The ratio of the primary dissociation rate is higher than the ensemble $S_{1/2}$, whereas the ratio with the secondary dissociation is lower. Because of restricted diffusion (*Berg, 1985*; *Nag and Dinner, 2006*), the apparent $K_D$ of a surface tethered molecule is always larger than the ensemble $S_{1/2}$. Therefore, we conclude that the primary high to low PIFE transition is the result of Reb1 dissociation. Furthermore, as we concluded earlier that the secondary low to high PIFE transition rate represents a Reb1–DNA structural transition, and should

have an associated high to low transition, we now conclude that the secondary high to low PIFE transitions are the result of Reb1–DNA structural transitions. Finally, our interpretation of the PIFE transitions is further confirmed by our observation that both primary rates were associated with 75% of the fluctuations and that both secondary rates were associated with 25% of the fluctuations. Overall, these results indicate that Reb1 mainly remains bound for about a second, while exhibiting occasional transitions to a long-lived bound state.

## Reb1 binds and dissociates from nucleosomes significantly slower than DNA

To compare Reb1 binding to nucleosomes versus DNA, we next measured Reb1-binding kinetics during its interaction with nucleosomes using single-molecule FRET (*Gibson et al., 2016*). We focused on the P8 nucleosomes as we would be able to compare these results to those of previous investigations of TF–nucleosome interactions (*Bernier et al., 2015*; *Li et al., 2005*; *Luo et al., 2014b*; *North et al., 2012*; *Simon et al., 2011*). The ensemble experiments (*Figures 1* and *2*) demonstrate that Reb1 binding to P8 nucleosomes can be detected by monitoring the change in FRET efficiency. To adapt this experiment for smTIRF, we reconstituted P8 nucleosomes with DNA that contained an additional 75-bp linker sequence on the side of the nucleosome opposite from the Reb1-binding site and the Cy3 fluorophore (*Figure 4A*). With these nucleosomes, we measured an ensemble FRET $S_{1/2\ \text{Reb1–smNuc P8 FRET}}$ of 2.2 ± 0.2 nM, very similar to that measured for 147-bp P8 nucleosomes (*Figure 4B*) and demonstrated that this DNA extension does not impact Reb1 interactions with nucleosomes.

We acquired 30 min FRET efficiency time series for more than 130 nucleosomes at separate Reb1 concentrations (2, 5, 10, 15 nM). Each nucleosome was immobilized on the microscope slide with a biotin–streptavidin linkage (*Figure 4A*) and fluctuates between high and low FRET states (*Figure 4C*). As we increased the Reb1 concentration, the FRET time series shifts to a larger fraction of time in the low FRET state (*Figure 4C*), indicating that Reb1 binds nucleosomes in a partially unwrapped low FRET state. We interpret the high FRET state as a fully wrapped nucleosome without bound Reb1, and the low FRET state as a partially unwrapped nucleosome with bound Reb1, as reported for other studies of TFs binding to nucleosomes (*Gibson et al., 2016*; *Luo et al., 2014a*; *Luo et al., 2014b*).

The cumulative sums of the unbound (high FRET) dwell times were fit best to single exponential distributions based on log-likelihood ratio tests (*Figure 4D*, *Figure 4—figure supplement 1A,B*). The rates from the exponential fits increased linearly with increasing Reb1 concentration, where the slope ($k_{\text{on Reb1–Nuc}}$ = 0.0006 ± 0.0001 s$^{-1}$ nM$^{-1}$) is the binding rate of Reb1. The rate of Reb1's binding to its site within the nucleosome relative to that in DNA is reduced by 53-fold. This is consistent with the occurrence of Reb1 binding within nucleosomes using the site exposure mechanism as similar reductions in binding rates have been observed for other TFs that employ this mechanism when binding nucleosomes (*Luo et al., 2014b*).

Log-likelihood ratio tests showed that the cumulative sums of the bound (low FRET) dwell times were best fitted to double exponential distributions (*Figure 4D*, *Figure 4—figure supplement 1A, B*). Both rates were independent of Reb1 concentration (*Figure 4E*, , *Figure 4—figure supplement 1C*) and implied Reb1 dissociation rates of $k_{\text{off Reb1–Nuc primary}}$ = 0.0044 ± 0.0005 s$^{-1}$ and $k_{\text{off Reb1–Nuc secondary}}$ = 0.07 ± 0.02 s$^{-1}$ (*Supplementary file 1* Table S2). The majority of bound events (>60%) belonged to the slower population (*Figure 4—figure supplement 1D*), suggesting that the slower rate is the primary mode for Reb1 dissociation. Interestingly, the rate of Reb1 dissociation from nucleosomes is ~130-fold lower than that of its dissociation from DNA, which is similar to the ~50-fold reduction in Reb1 binding to nucleosomes relative to DNA. This reduction in dissociation rate compensates for the reduction in binding rate and results in a similar Reb1 binding affinities for its sites within nucleosomes and DNA. This result suggests that Reb1 interacts with partially unwrapped nucleosomes differently than other TFs, such as Gal4 and LexA, which exhibit ~1000-fold acceleration in dissociation rates from nucleosomes compared to DNA (*Luo et al., 2014b*).

The ratio of the dissociation rate to the binding rate is the apparent dissociation constant, $K_D$, which can be compared to the ensemble $S_{1/2}$ measurements. As binding rates are known to be influenced by restricted diffusion resulting from surface tethering (*Berg, 1985*; *Nag and Dinner, 2006*), we compared the ratio of single-molecule apparent $K_D$s for binding to nucleosomes and DNA to the ratio of ensemble $S_{1/2}$s for binding nucleosomes and DNA (*Figure 4—figure supplement 1E*,

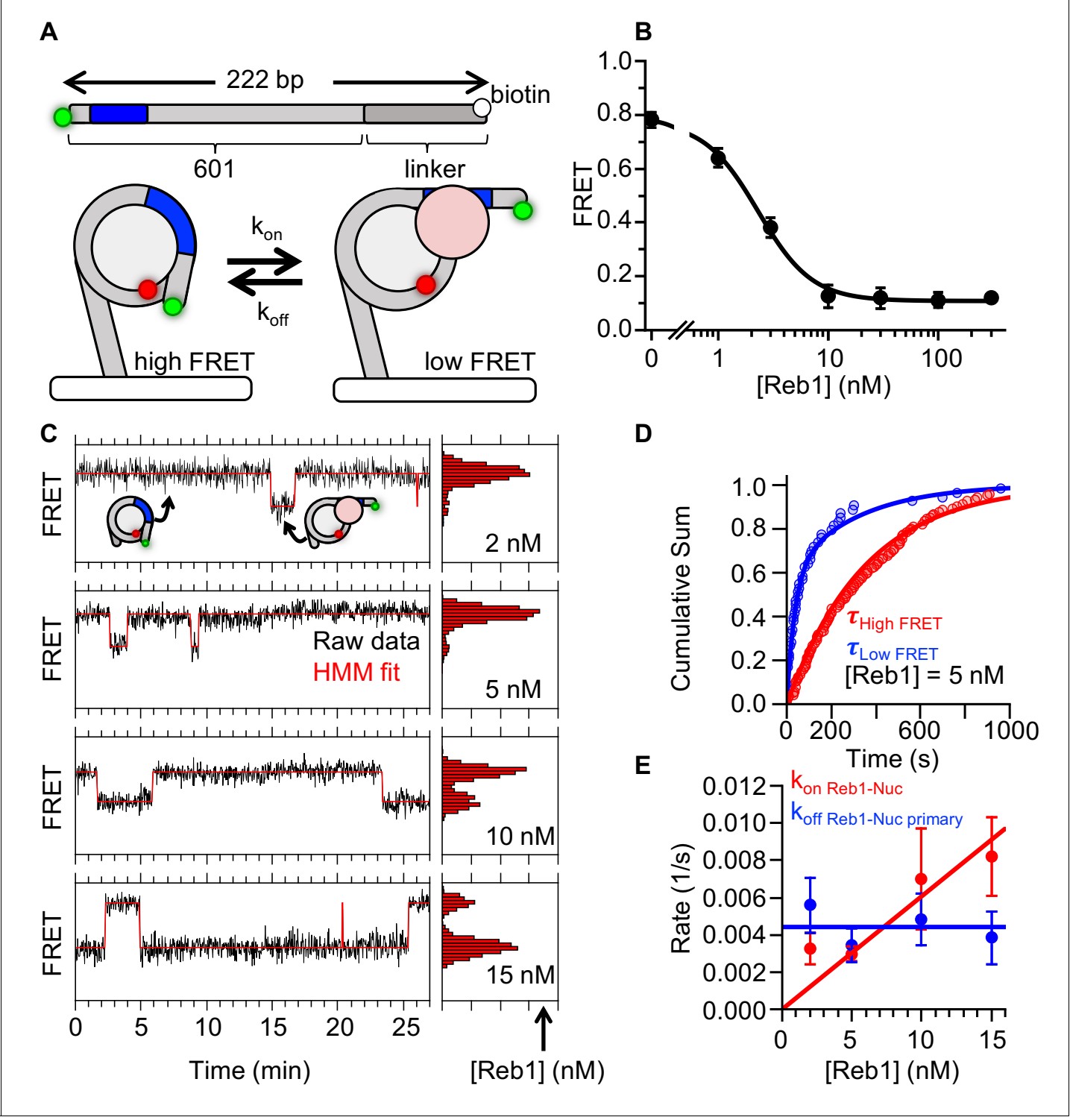

**Figure 4.** Reb1 binds and dissociates from nucleosomes significantly slower than DNA. (**A**) smFRET P8 nucleosomes are tethered to the microscope surface through an additional 75 bp of DNA extending out of the nucleosome opposite to Cy3 and the Reb1-binding site. The octamer was labeled with Cy5 at H2A(K119C). Reb1 binding traps the nucleosome in a low FRET state. (**B**) Ensemble FRET titration of Reb1 with smFRET nucleosomes. The titration fits to a binding isotherm with an $S_{1/2\ Reb1-smNuc\ P8\ FRET}$ = 2.2 ± 0.2. This value is similar to that for titrations with nucleosomes containing 147-bp DNA (***Figure 2B***). (**C**) Example time traces of single nucleosomes at four separate Reb1 concentrations, which are fitted with a two-state Hidden-Markov Model. As the Reb1 concentration increases, the immobilized nucleosome shifts to the low FRET state. (**D**) Cumulative sums of dwell times in the high FRET (red) and low FRET states (blue), which fit to single and double exponentials, respectively. The Reb1 concentration is 5 nM. (**E**) The

*Figure 4 continued on next page*

*Figure 4 continued*

primary Reb1-binding (red) and -dissociation (blue) rates for increasing Reb1 concentrations. The dissociation rates are constant with an average rate of $k_{off\ Reb1-Nuc\ primary} = 0.0044 \pm 0.0005\ s^{-1}$, whereas the binding rates fit to a line with a slope that equals the overall binding rate of $k_{on\ Reb1-Nuc\ primary} = 0.0006 \pm 0.0001\ s^{-1}\ nM^{-1}$.

DOI: https://doi.org/10.7554/eLife.43008.014

The following figure supplement is available for figure 4:

**Figure supplement 1.** Analysis of Reb1–nucleosome single-molecule binding experiments.

DOI: https://doi.org/10.7554/eLife.43008.015

*Supplementary file 1* Table S3). This will largely remove the impact of the restricted diffusion because the on rates will be impacted similarly for tethered nucleosomes and DNA. When using dominant rates, we found that the relative changes in binding affinity between nucleosomes and DNA are in close agreement for single-molecule and ensemble experiments. This strongly suggests that $k_{off\ Reb1-Nuc\ primary}$ is the dominant determinate of the dissociation rate in the Reb1–nucleosome interaction in solution and that surface tethering does not impact the measured dissociation rates.

## Cbf1 also binds and dissociates from nucleosomes significantly slower than from DNA

Recent work established that six TFs, including Reb1, are mainly responsible for NDR generation in *S. cerevisiae* (*Yan et al., 2018*). To determine whether slow dissociation rates for binding sites within nucleosomes might be a general property of these factors, we performed experiments to examine the binding of Cbf1 to DNA and nucleosomes. Cbf1 is another member of the group of TFs responsible for NDR generation in *S. cerevisiae*. It exhibits little structural similarity to Reb1; it contains only one DNA-binding domain (myc-like), is significantly smaller (39 kDa) than Reb1, and binds to DNA as a dimer (*Wieland et al., 2001*). However, like that of Reb1, its N-terminus is negatively charged and predicted to be unstructured (*Rost et al., 2004*).

EMSA and ensemble PIFE measurements indicated that Cbf1 tightly binds DNA ($S_{1/2\ Cbf1-DNA\ PIFE} = 1.3 \pm 0.3\ nM$) and that this binding can be detected through Cy3 PIFE (*Figure 5A*, *Figure 5—figure supplement 1A*, *Supplementary file 1* Table S1). We then performed smPIFE experiments and detected fluctuations representative of binding as we did for Reb1 (*Figure 5C*). The dwell time cumulative sums followed double exponential distributions (*Figure 5D*, *Figure 5—figure supplement 1C–F*, *Supplementary file 1* Table S2). As for Reb1, the primary low to high PIFE transition was concentration dependent, implying that it is due to Cbf1-binding events ($k_{on\ Cbf1-DNA\ primary} = 0.025 \pm 0.006\ s^{-1}\ nM^{-1}$), whereas the secondary low PIFE dwell times are concentration independent and are likely to result from Cbf1–DNA structural transitions ($k_{on\ Cbf1-DNA\ secondary} = 0.024 \pm 0.003\ s^{-1}$). In addition, we detected two separate high to low PIFE transition rates ($k_{off\ Cbf1-DNA\ primary} = 0.30 \pm 0.05\ s^{-1}$, $k_{off\ Cbf1-DNA\ secondary} = 0.034 \pm 0.004\ s^{-1}$), where the primary rate is due to Cbf1 dissociation and the secondary rate is due to Cbf1–DNA structural transition.

We then used both EMSA and ΔFRET to detect Cbf1 binding to Cy3-Cy5-labeled nucleosomes with the Cbf1 binding site at the P8 position. We used the same Cy3- and Cy5-labeled positions as were used for the Reb1 FRET measurements (*Figure 2A*). Titrating Cbf1 resulted in a significant reduction in FRET with an $S_{1/2\ Cbf1-Nuc\ FRET} = 12 \pm 2\ nM$, which is similar to the EMSA measurements and indicates that FRET can be used to measure Cbf1 binding (*Figure 5B*, , *Figure 5—figure supplement 1B*, *Supplementary file 1* Table S1). Comparison of these ensemble measurements reveals a ~10-fold lower binding affinity of Cbf1 to P8 nucleosomes as compared with that for binding to naked DNA. We then used smFRET to determine the kinetic rates of Cbf1 binding to and dissociating from P8 nucleosomes (*Figure 5E*). The cumulative sums were best fitted with a single exponential distribution ($k_{on\ Cbf1-Nuc} = 0.00021 \pm 0.00002\ s^{-1}\ nM^{-1}$ and $k_{off\ Cbf1-Nuc} = 0.0111 \pm 0.0007 s^{-1}$) (*Figure 5F*, *Figure 5—figure supplement 1C–F*). As for Reb1, using the dominant rates from these measurements, we determined that the relative change in the affinity of Cbf1-binding to DNA or to nucleosomes is consistent between single-molecule and ensemble measurements (*Figure 5—figure supplement 1G*, *Supplementary file 1* Table S3).

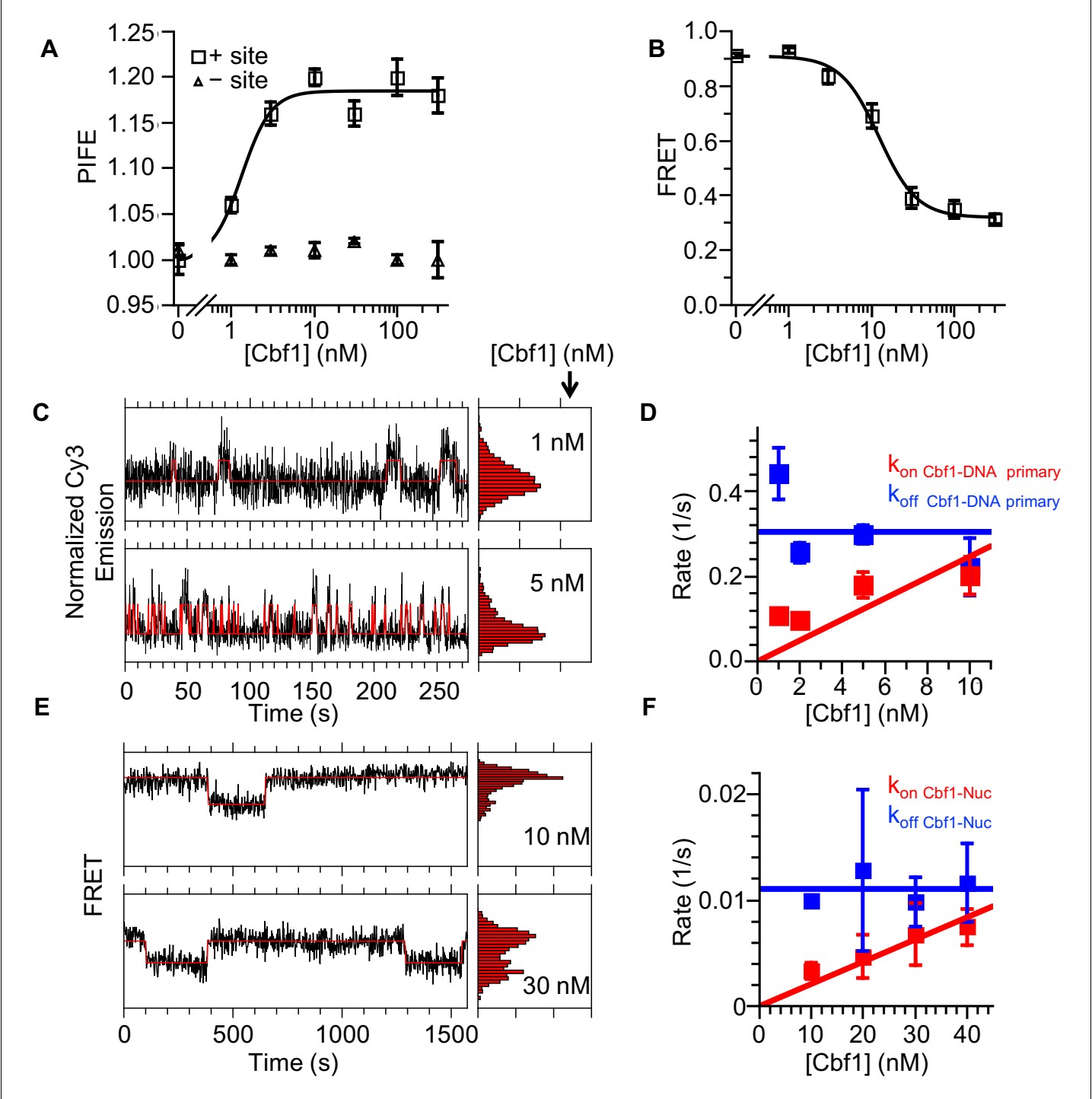

**Figure 5.** Cbf1 also binds and dissociates from nucleosomes significantly slower than from DNA. (**A**) Ensemble PIFE measurement of a Cbf1 titration with a 94-bp DNA with and without the Cbf1-binding sites 1 bp from the 5′ end and Cy3 labeled. The normalized PIFE fits to a binding isotherm with an $S_{1/2\ Cbf1-DNA\ PIFE} = 1.3 \pm 0.3$ nM. Without the binding site, the Cy3 emission does not change, demonstrating that the observed PIFE is due to site-specific Cbf1 binding. (**B**) Cbf1 titration with Cy3-Cy5 labeled nucleosomes with the Cbf1 site at P8. The FRET fits to a binding isotherm with an $S_{1/2\ Cbf1-smNuc\ FRET} = 12.3 \pm 1.6$ nM. (**C**) Example time traces of single DNA molecules for two separate Cbf1 concentrations, where the black lines are the Cy3 fluorescence and the red lines are the two-state Hidden Markov Model fits. As the Cbf1 concentration increases, the immobilized DNA molecules shift to the high PIFE state. (**D**) The Cbf1–DNA primary binding and dissociation rates for increasing concentrations of Cbf1. These were determined from cumulative sums of Cbf1–DNA high PIFE and low PIFE dwell times that were fitted to double exponentials. The primary dissociation kinetics (blue) were constant with an average of $k_{off\ Cbf1-DNA\ primary} = 0.30 \pm 0.05\ s^{-1}$, while the primary binding kinetics (red) fit to a line with a slope that equals the

*Figure 5 continued on next page*

*Figure 5 continued*

overall binding rate of $k_{\text{on Cbf1–DNA primary}} = 0.025 \pm 0.006$ s$^{-1}$ nM$^{-1}$. (E) Example time traces of single nucleosomes with two separate Cbf1 concentrations, where the black lines are the FRET efficiency data and the red lines are the two-state Hidden Markov Model fits. As the Cbf1 concentration increases, the immobilized nucleosome shift to the low FRET state. (F) The Cbf1–nucleosome binding and dissociation rates for increasing concentrations of Cbf1. These were determined from cumulative sums of Cbf1–nucleosome low FRET and high FRET dwell times that were fitted to single exponentials. The dissociation kinetics (blue) were constant with an average of $k_{\text{off Cbf1–Nuc}} = 0.0111 \pm 0.0007$ s$^{-1}$, whereas the binding kinetics (red) fit to a line with a slope that equals the overall binding rate of $k_{\text{on Cbf1–Nuc}} = 0.00021 \pm 0.00002$ s$^{-1}$ nM$^{-1}$.

DOI: https://doi.org/10.7554/eLife.43008.016

The following figure supplement is available for figure 5:

**Figure supplement 1.** Characterizing Cbf1 interactions with DNA and nucleosomes.
DOI: https://doi.org/10.7554/eLife.43008.017

As for Reb1, we detect a slower rate (120-fold) for Cbf1-binding to nucleosomes than for its binding to DNA, which indicates that Cbf1 also gains access to nucleosomes via the site exposure mechanism. Interestingly, the primary rate of Cbf1 dissociation from nucleosomes is ~25-fold lower than that from DNA, a comparison that is qualitatively similar to that for Reb1 and in stark contrast to the orders of magnitude increase in corresponding dissociation rates observed for the Gal4 and LexA TFs (*Luo et al., 2014b*). The decreased rate of Cbf1–nucleosome dissociation partially compensates for the decreased binding rate and explains why Cbf1 binds to its site within nucleosomes with an affinity that is only about 10-fold weaker than that for its binding to DNA, as compared to the orders of magnitude decrease in occupancy observed for both Gal4 and LexA (*Liang et al., 1996*; *Luo et al., 2014b*; *Polach and Widom, 1995*). Combined, these results for Cbf1 and Reb1 indicate that, like PFs in higher eukaryotes, these TFs have high affinity for nucleosomal substrates, and that they achieve this high affinity by reducing their dissociation rates to compensate for their reduced binding rates.

## The N-terminal tail of Reb1 does not contribute significantly to the pioneer property of Reb1

The group of six budding yeast TFs that can efficiently create NDRs, including Reb1 and Cbf1 (*Yan et al., 2018*), are all highly acidic as compared to proteins that do not have such activity (*Figure 6—figure supplement 1A*). More specifically, both Reb1 and Cbf1 contain long acidic N-terminal tails that could form favorable electrostatic interactions with the basic histone surface that is exposed in a partially unwrapped nucleosome. This could prevent nucleosome rewrapping and significantly reduce their dissociation rates. To investigate whether the acidic N-terminal tail is important for Reb1–nucleosome binding, we generated a Reb1 truncation mutant, Reb1-ΔN, in which the first 395 amino acids are removed (*Figure 6A*). We then used both EMSAs and ensemble fluorescence to quantify Reb1-ΔN binding to both DNA and P8 nucleosomes (*Figure 6B and C*, *Figure 6—figure supplement 1B*). We found that Reb1-ΔN binds its target site within DNA with a 1.5-fold higher affinity than to its site within P8 nucleosomes (*Figure 6F*; $S_{\text{1/2 Reb1-ΔN–DNA PIFE}} = 7.8 \pm 0.5$ nM; $S_{\text{1/2 Reb1-ΔN–Nuc P8 FRET}} = 11.8 \pm 0.9$ nM). This value of relative binding within DNA and P8 nucleosomes is only mildly higher than for the full-length Reb1 (2.1-fold), indicating that the acidic N-terminal tail is not required for Reb1 to target nucleosomes similarly to DNA.

We then used single-molecule fluorescence to investigate the Reb1-ΔN binding and dissociation rates (*Figure 6D,E*, *Figure 6—figure supplement 2*). We found that the rate of Reb1-ΔN binding to P8 nucleosomes ($0.0003 \pm 0.00001$ s$^{-1}$ nM$^{-1}$) was about 2-fold slower than the rate of binding of full-length Reb1 to this site, while the primary dissociation rate of Reb1-ΔN from P8 nucleosomes ($0.0044 \pm 0.0008$ s$^{-1}$) was the same as that for full-length Reb1. This result confirms the ensemble studies described above and implies that Reb1's acidic N-terminal domain is not responsible for the 130-fold decrease in the rate of dissociation from nucleosomes. Therefore, it appears that Reb1's C-terminal DNA-binding domain is responsible for its pioneer activity.

## Reb1 slowly exchanges in vivo

To further investigate our observation that Reb1 can function as a PF, we carried out FRAP measurements. Previous FRAP measurements of mammalian green fluorescent protein (GFP)-tagged PFs show that they exchange with a characteristic recovery time that is significantly slower than

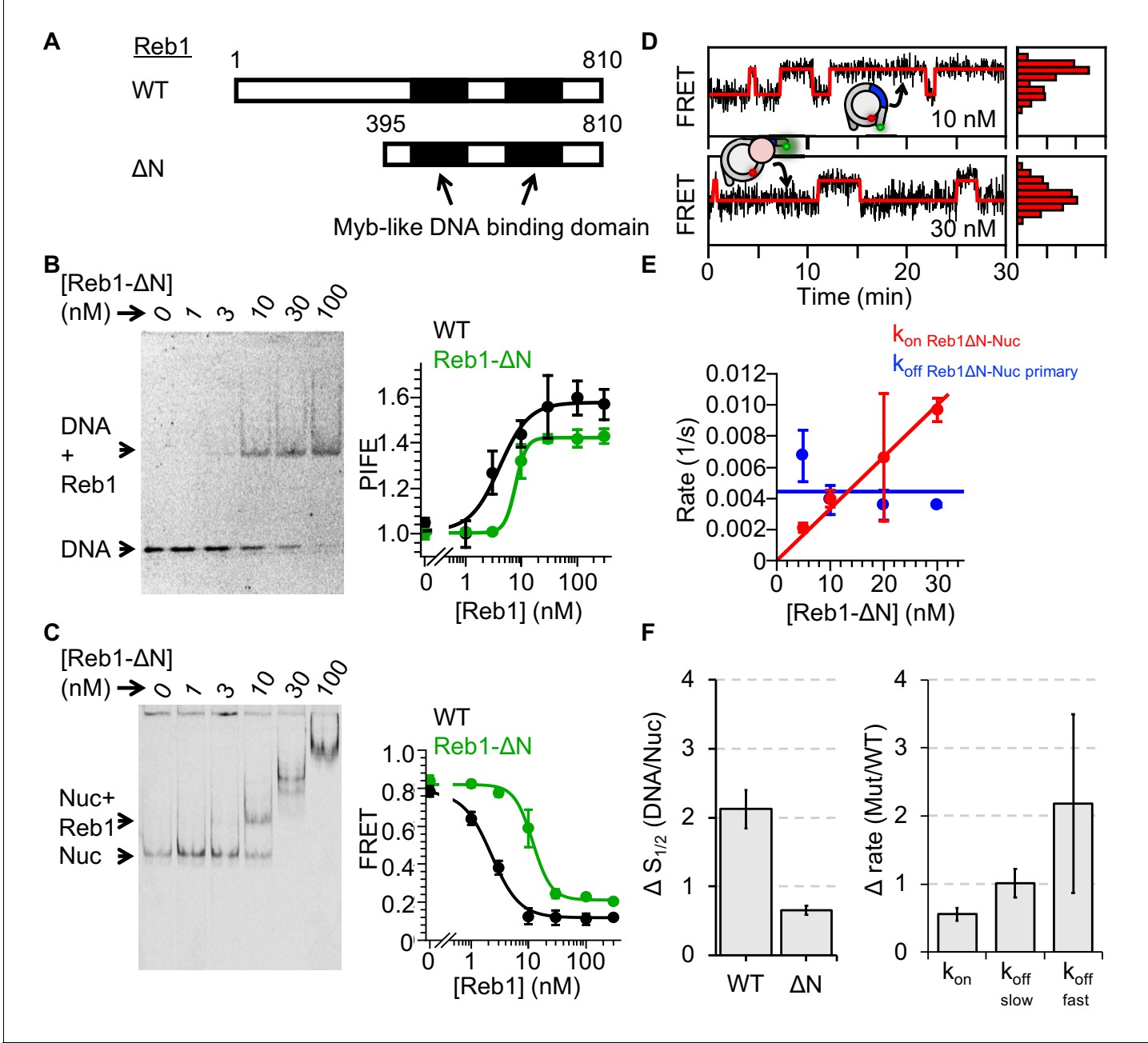

**Figure 6.** Reb1ΔN binds and dissociates similarly to WT Reb1. (A) Schematic comparison of WT Reb1 and the deletion variant Reb1-ΔN, which is comprised of residues 395–810. (B) Cy3 image of the EMSA of Reb1-ΔN binding to the 25-bp DNA molecule ($S_{1/2\ Reb1\Delta N–DNA\ EMSA}$ = 12.8 ± 1.2 nM). In addition, Reb1-ΔN titration with the smPIFE DNA results in a Cy3 emission increase of ~1.4-fold and fits to a binding isotherm with an $S_{1/2\ Reb1\Delta N–DNA\ PIFE}$ = 7.8 ± 0.5 nM. (C) Cy5 image of Reb1-ΔN binding to smFRET nucleosomes containing the P8 Reb1-binding site ($S_{1/2\ Reb1\Delta N\ -Nuc\ P8\ EMSA}$ = 8.5 ± 0.5 nM). In addition, ensemble FRET measurements with these nucleosomes fit to a binding isotherm with an $S_{1/2\ Reb1\Delta N–Nuc\ P8\ FRET}$ = 11.8 ± 0.9 nM. (D) Example time traces of single nucleosomes for two separate Reb1-ΔN concentrations, which are fitted to a two-state Hidden-Markov Model. As the Reb1 concentration increases, the immobilized nucleosome shifts to the low FRET state. (E) The primary Reb1-binding (red) and -dissociation (blue) rates for increasing Reb1-ΔN concentrations. The dissociation rates are constant with an average rate of $k_{off\ Reb1\Delta N–Nuc\ primary}$ = 0.0044 ± 0.0008 s$^{-1}$, whereas the binding rates fit to a line with a slope that equals the overall binding rate of $k_{on\ Reb1\Delta N–Nuc\ primary}$ = 0.0003 ± 0.00001 s$^{-1}$ nM$^{-1}$.
DOI: https://doi.org/10.7554/eLife.43008.018

The following figure supplements are available for figure 6:

**Figure supplement 1.** Strong nucleosome displacing factors are characterized by their overall negative charge.
DOI: https://doi.org/10.7554/eLife.43008.019

**Figure supplement 2.** Characterizing Reb1ΔN interactions with nucleosomes.

*Figure 6 continued on next page*

*Figure 6 continued*

DOI: https://doi.org/10.7554/eLife.43008.020

that of other TFs in the nuclei (*Plachta et al., 2011*; *Sekiya et al., 2009*). We carried out FRAP measurements of endogenously expressed GFP-tagged Reb1 in *S. cerevisiae*, and observed that Reb1 fluorescence recovers with a half life of 25.8 ± 2.5 s (*Figure 7*), which is similar to the exchange times observed for the mammalian PF, FoxA. We could not get high-quality FRAP data for Cbf1, which is less abundant than Reb1.

For comparison, we performed FRAP measurements for three additional endogenously expressed GFP-tagged proteins that interact with chromatin: histone H3, Sth1, and Nhp6A. These factors were chosen because they are proteins that are abundant in the nucleus and are expected to have different levels of chromosome engagement. H3 is stably integrated into chromatin and has been previously reported to exchange on the hour time scale in mammalian cells (*Kimura and Cook, 2001*). Sth1 is a subunit of the nucleosome remodeling complex RSC, which has strong nucleosome interactions but which interacts with chromatin transiently (*Erdel et al., 2010*; *Yen et al., 2012*). Nhp6A is a high-mobility-group protein that binds to DNA with low sequence-specificity (*Stillman, 2010*), and its mammalian homolog, HMGB1, was shown to have very rapid FRAP recovery rates (*Sekiya et al., 2009*). We found the FRAP half-time of H3 to be much longer than the minute time scale, the length of our experiment. By contrast, the half-lives for Sth1 and Nhp6A were 7.8

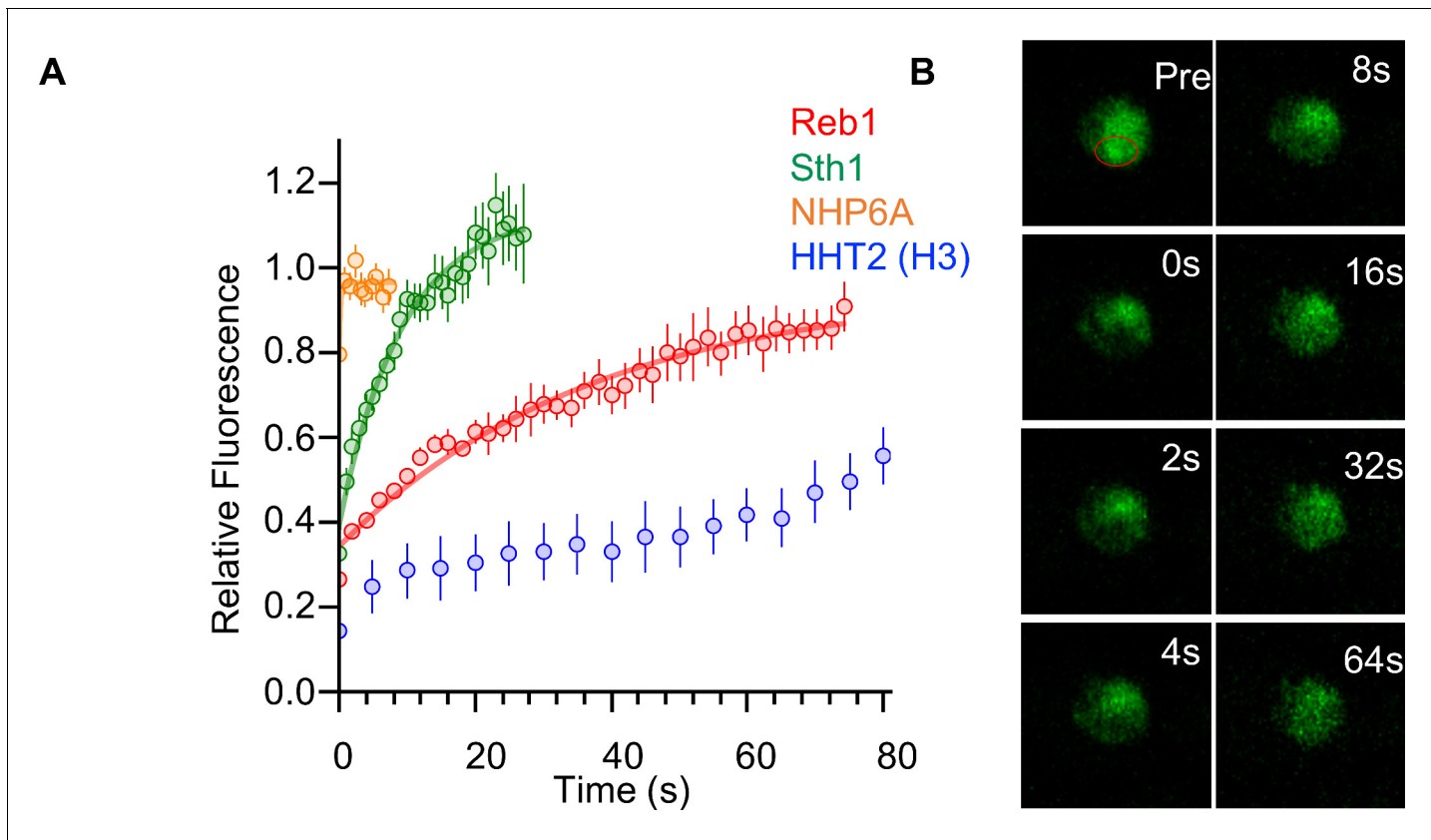

**Figure 7.** Reb1 slowly exchanges in vivo. (**A**) Recovery curves for Reb1 (red), HHT2 (Blue), Sth1 (green), and NHP6A (orange) after photobleaching. Reb1 $\tau_{1/2}$ = 25.8 ± 2.5 s, Sth1 $\tau_{1/2}$ = 7.8 ± 0.7 s, NHP6A $\tau_{1/2}$ = 0.2 ± 0.1 s. (**B**) Fluorescence images of GFP-labeled Reb1 during a FRAP experiment. The bleached region is indicated with a red circle.

DOI: https://doi.org/10.7554/eLife.43008.021

The following figure supplement is available for figure 7:

**Figure supplement 1.** Fluorescence recovery of non-PF proteins in a yeast nucleus.

DOI: https://doi.org/10.7554/eLife.43008.022

s and <1 s, respectively (*Figure 7—figure supplement 1*). This shows that Reb1 exchanges faster than the chromatin-forming protein histone H3, but slower than other transcription-regulatory proteins that transiently interact with chromatin. This result provides additional evidence to show that Reb1 functions in vivo similarly to mammalian PFs, i.e. it exchanges more slowly than other transcription-regulatory complexes.

## Discussion

We combined ensemble, single-molecule and live-cell fluorescence studies to investigate mechanistically how the budding yeast TFs Reb1 and Cbf1 interact with nucleosomal templates. We find that, like PFs, Reb1 and Cbf1 occupy sites within the nucleosome with affinities similar to those with which they occupy to naked DNA. These factors invade the nucleosome and trap it in a partially unwrapped state using the site exposure model, which results in a significant reduction in the binding rate (*Li et al., 2005*; *Tims et al., 2011*). Interestingly, Reb1 completely and Cbf1 partially compensates for this *binding* rate reduction by reducing their *dissociation* rates (*Figure 8A*). This dissociation rate compensation mechanism explains how a TF can have similar affinities on naked and nucleosomal DNA (*Figure 8B*), as has been proposed for the human PF FoxA (*Cirillo and Zaret, 1999*).

Although yeast and human PFs bind nucleosomes and DNA with similar affinities, their impacts on nucleosome structure and dynamics may be distinct. For example, Reb1 and Cbf1 can trap nucleosomes in an unwrapped state, but human PFs do not appear to influence nucleosomal DNA unwrapping. FoxA traps mobile nucleosomes in distinct positions along the DNA (*Cirillo et al., 2002*; *Cirillo and Zaret, 1999*) and competes with linker histones (*Cirillo et al., 2002*), whereas Oct4 and Sox2 do not appear to disrupt nucleosomes upon binding to the DNA entry-exit region (*Soufi et al., 2015*). The dissociation rate compensation mechanism, which allows PFs to target their DNA sites within nucleosomes efficiently, appears to be conserved from yeast to humans, whereas the impact of these factors on the structure and binding of nucleosomes is much more diverse.

The specificity of Reb1 for its site near the edge of the nucleosome is 22-fold (*Figure 1B*), where specificity is defined as the fold-enhancement in binding affinity when a binding site is included the nucleosome. This is less than that of traditional TFs, such as the bacterial TF LacI (*Lin and Riggs, 1975*) and the eukaryotic TF Gal4 (*Liang et al., 1996*). However, the most extensively characterized pioneer factor, FoxA, has only a 2-fold difference in specificity (*Sekiya et al., 2009*), indicating that the Reb1 specificity is not out of line with that of other pioneer factors. The mechanisms of how TFs find their target sites in vivo are not yet resolved, but a potential model for Reb1 is that it is kinetically regulated; the bound-state residence time instead of the bound-state probability is key for Reb1 function. The residence time of Reb1 on its specific sites, especially the sites within the edge of the nucleosomes, is likely to be much longer than that on nonspecific sites. The long residence time may be required for Reb1 to recruit co-activators, such as chromatin remodelers and histone modifying complexes. In addition, Reb1's target sites in vivo tend to be located in promoters, where many other regulatory factors bind. The cooperative binding of Reb1 and these factors may further enhance Reb1's binding specificity.

Our results on Reb1 and Cbf1 dissociation rates are strikingly different to previous results on TFs such as Gal4 and LexA, whose dissociation rates from nucleosomes are much higher than those from DNA. This acceleration has been proposed to be the result of both competition between nucleosome rewrapping and the maintenance of TF partially bound states and disfavorable TF–nucleosome interactions such as steric clash (*Chen and Bundschuh, 2014*). The reason why Reb1 and Cbf1 dwell much longer on their nucleosomal sites is still not clear. The human PF FoxA stabilizes its binding to nucleosomes by contacting H3 via its C-terminus (*Cirillo et al., 2002*), so we suspect that a similar mechanism may also stabilize Reb1 and Cbf1 binding. Interestingly, we find that the long acidic N-terminal tail of Reb1 does not contribute to the dissociation rate kinetic compensation mechanism. This indicates that the DNA-binding domain of Reb1 preferentially interacts with its site within the DNA entry-exit region of the nucleosome. Furthermore, because the acidic N-terminal domain of Reb1 does not influence binding to nucleosomes, it appears to have a different function such as recruiting transcription activators such as chromatin remodelers and histone-modifying complexes.

**A**

| TF | $k_{off\ DNA}$ (s$^{-1}$) | $k_{off\ Nuc}$ (s$^{-1}$) | $k_{off\ DNA}$ / $k_{off\ Nuc}$ | $k_{on\ DNA}$ (s$^{-1}$ nM$^{-1}$) | $k_{on\ Nuc}$ (s$^{-1}$ nM$^{-1}$) | $k_{on\ DNA}$ / $k_{on\ Nuc}$ |
|---|---|---|---|---|---|---|
| Reb1 | $0.58 \pm 0.08$ | $(4.4 \pm 0.5)$ $\times 10^{-3}$ | $130 \pm 20$ | $(3.2 \pm 0.3)$ $\times 10^{-2}$ | $(6 \pm 1)$ $\times 10^{-4}$ | $53 \pm 1$ |
| Cbf1 | $0.30 \pm 0.05$ | $(1.1 \pm 0.1)$ $\times 10^{-2}$ | $27 \pm 5$ | $(2.5 \pm 0.6)$ $\times 10^{-2}$ | $(2.1 \pm 0.2)$ $\times 10^{-4}$ | $120 \pm 30$ |
| Gal4 | $\ll 5 \times 10^{-4}$ | $(2.0 \pm 0.1)$ $\times 10^{-2}$ | $\ll 0.025$ | ND | $0.40 \pm 0.02$ | ND |
| LexA | $(3.4 \pm 0.2)$ $\times 10^{-3}$ | $3.3 \pm 0.6$ | $0.001 \pm 0.0002$ | $(5 \pm 1)$ $\times 10^{-2}$ | $(9 \pm 2)$ $\times 10^{-5}$ | $500 \pm 200$ |

**B**

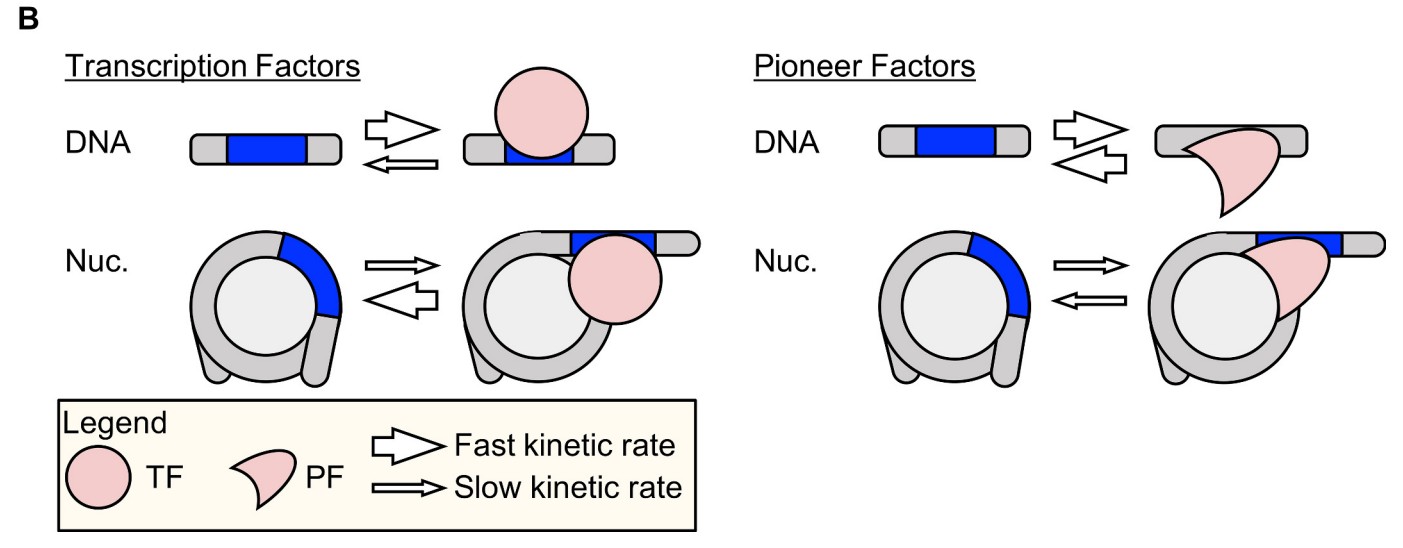

**Figure 8.** Dissociation rate compensation mechanism for yeast pioneer TFs. (**A**) Table of Reb1, Cbf1, Gal4 and LexA (*Luo et al., 2014b*) binding and dissociation rates with DNA and P8 nucleosomes. (**B**) The dissociation rate compensation mechanism. (Left) For traditional TFs, nucleosomes decrease TF binding rates and increase TF dissociation rates, which can reduce the overall TF affinity by orders of magnitude. (Right) Like TFs, nucleosomes have lower PF binding rates than do naked DNA, but the PF dissociation rate from nucleosomes is lower than that from DNA. This compensates for the reduced PF binding rate so that the overall PF affinity is similar for nucleosomes and DNA, and allows PFs to trap nucleosomes in partially unwrapped states efficiently.

DOI: https://doi.org/10.7554/eLife.43008.023

Interestingly, Reb1 binds to position P3 with 3.3-fold less affinity than to P8, in spite of the fact that the P3 position is five base pairs closer to the edge of the nucleosome and therefore more accessible. There are two potential non-exclusive reasons for this difference. First, the DNA will not unwrap as much for Reb1 binding to P3, so less histone octamer surface should be exposed. If Reb1 achieves a higher binding affinity to nucleosomes by interacting with the exposed histone surface, this reduction in exposed histone octamer surface would reduce the affinity of Reb1 at the P3 position. Second, the P3 position is shifted by 5 bp, which will result in an 180 degree rotation of the binding site relative to the histone octamer. This could orient the binding region of Reb1 away from the histone octamer, so that it is unable to interact with the exposed histone octamer surface, thereby reducing the overall affinity of this site at P3 relative to P8.

Our observation that Reb1 binding to P18 nucleosomes does not trap the nucleosome in an unwrapped state indicates that the mode of Reb1 binding to this site, which is further into the nucleosome, is distinct from binding very near the edge of the nucleosome at P3 and P8. Although Reb1 affinity at position P18 is 7.5-fold lower than that at position P8, Reb1's affinity for P18 is still 2.9-fold greater than its affinity for nucleosomes without the Reb1 site. This indicates that the Reb1 binding remains site-specific in a manner similar to that of the human pioneer factor FoxA (*Sekiya et al., 2009*). Although it remains unclear how shifting the site by 10 bp changes the Reb1-binding mode, previous studies provide insight. At position P18, the Reb1-binding site extends 27 bp into the nucleosomes. Given the ~90 kDa size of Reb1, the amount of DNA that is required to unwrap so that Reb1 can fully recognize the P18 binding site could extend significantly further into the nucleosomes than 27 base pairs. There are strong DNA–histone contacts that are located 25 to 35 base pairs into the nucleosome (*Hall et al., 2009*) and these cost 5 $K_B$T of free energy to unwrap (*Forties et al., 2011*). This implies that the unwrapping probability is reduced by at least 150-fold. This significant increase in nucleosome unwrapping free energy could mean that Reb1 preferentially binds part of the P18 target site within a fully wrapped nucleosome, as has been proposed for other pioneer factors such as Oct4 (*Soufi et al., 2015*).

The findings described above have significant implications for explaining how certain TFs generate NDRs in vivo. A previous study proposed that TFs in *S. cerevisiae* can invade into nucleosomes passively by trapping transiently exposed naked DNA during replication or other histone-turnover events (*Yan et al., 2018*). However, this does not exclude the possibility that some TFs can directly bind and invade into nucleosomes. Our observation that Reb1 and Cbf1 stably engage nucleosome provides support for the latter mechanism. Given that these two factors do not evict histones in vitro (nor do other PFs), we do not think it is likely that these factors cause spontaneous nucleosome disassembly in vivo. Instead, the long dwell time of these factors on nucleosomes may allow time to recruit other factors, such as histone chaperones or nucleosome remodelers, to establish NDRs. Future in vivo and in vitro measurements are needed to investigate this further.

## Materials and methods

**Key resources table**

| Reagent type (species) or resource | Designation | Source or reference | Identifiers | Additional information |
|---|---|---|---|---|
| Strain, strain background (*S. cerevisiae*) | BY4741 | (*Huh et al., 2003*) | MATa his3Δ1 leu2Δ0 met15Δ0 ura3Δ0 | |
| Strain, strain background (*S. cerevisiae*) | BY4742 | (*Huh et al., 2003*) | MATα his3Δ1 leu2Δ0 lys2Δ0 ura3Δ0 | |
| Strain, strain background (*S. cerevisiae*) | BY4741-Reb1 | Constructed in this study | MATa his3Δ1 leu2Δ0 met15Δ0 ura3Δ0 REB1–GFP::His3MX | See 'Materials and methods: FRAP assay' |
| Strain, strain background (*S. cerevisiae*) | BY4741-HHT2 | Constructed in this study | MATa his3Δ1 leu2Δ0 met15Δ0 ura3Δ0 HHT2 –GFP::His3MX | See 'Materials and methods: FRAP assay' |
| Strain, strain background (*S. cerevisiae*) | BY4741–Sth1 | Constructed in this study | MATa his3Δ1 leu2Δ0 met15Δ0 ura3Δ0 STH1 –GFP::His3MX | See 'Materials and methods: FRAP assay' |
| Strain, strain background (*S. cerevisiae*) | BY4741–NHP6A | Constructed in this study | MATa his3Δ1 leu2Δ0 met15Δ0 ura3Δ0 NHP6A –GFP::His3MX | See 'Materials and methods: FRAP assay' |

*Continued on next page*

*Continued*

| Reagent type (species) or resource | Designation | Source or reference | Identifiers | Additional information |
|---|---|---|---|---|
| Strain, strain background (*S. cerevisiae*) | BY4742–Reb1 | Constructed in this study | *MATα his3Δ1 leu2Δ0 lys2Δ0 ura3Δ0 REB1 –GFP::His3MX* | See 'Materials and methods: FRAP assay' |
| Strain, strain background (*S. cerevisiae*) | BY4742-HHT2 | Constructed in this study | *MATα his3Δ1 leu2Δ0 lys2Δ0 ura3Δ0 HHT2 –GFP::His3MX* | See 'Materials and methods: FRAP assay' |
| Strain, strain background (*S. cerevisiae*) | BY4742–Sth1 | Constructed in this study | *MATα his3Δ1 leu2Δ0 lys2Δ0 ura3Δ0 Sth1– GFP::His3MX* | See 'Materials and methods: FRAP assay' |
| Strain, strain background (*S. cerevisiae*) | BY4742 –NHP6a | Constructed in this study | *MATα his3Δ1 leu2Δ0 lys2Δ0 ura3Δ0 NHP6A –GFP::His3MX* | See 'Materials and methods: FRAP assay' |
| Strain, strain background (*S. cerevisiae*) | BY4743–Reb1 | Constructed in this study | *MATa/α his3Δ1/his3Δ1 leu2Δ0/leu2Δ0 LYS2/lys2Δ0 met15Δ0/MET15 ura3Δ0/ura3Δ0 REB1–GFP::His3MX/ Factor-GFP:: His3MX* | See 'Materials and methods: FRAP assay' |
| Strain, strain background (*S. cerevisiae*) | BY4743–HHT2 | Constructed in this study | *MATa/α his3Δ1/his3Δ1 leu2Δ0/leu2Δ0 LYS2/lys2Δ0 met15Δ0/MET15 ura3Δ0/ura3Δ0 HHT2–GFP::His3MX/ Factor-GFP:: His3MX* | See 'Materials and methods: FRAP assay' |
| Strain, strain background (*S. cerevisiae*) | BY4743–Sth1 | Constructed in this study | *MATa/α his3Δ1/his3Δ1 leu2Δ0/leu2Δ0 LYS2/lys2Δ0 met15Δ0/MET15 ura3Δ0/ura3Δ0 Sth1–GFP::His3MX/ Factor-GFP:: His3MX* | See 'Materials and methods: FRAP assay' |
| Strain, strain background (*S. cerevisiae*) | BY4743–NHP6A | Constructed in this study | *MATa/α his3Δ1/his3Δ1 leu2Δ0/leu2Δ0 LYS2/lys2Δ0 met15Δ0/MET15 ura3Δ0/ura3Δ0 NHP6A–GFP::His3MX/ Factor-GFP:: His3MX* | See 'Materials and methods' |

## Preparation of Reb1

Reb1 was cloned into pHIS8 and expressed/purified as previously described (*Krietenstein et al., 2016*). The Reb1 N-terminal truncation variant, 'Reb1-ΔN', was prepared via site-directed mutagenesis (Agilent Technologies) of the Reb1-WT plasmid using primers that anneal partially to the region immediately upstream of the TSS and partially to the region of the plasmid that codes for the C-terminus of the protein beginning at residue 396. Reb1 and Reb1-ΔN were expressed in *Escherichia coli* BL21(DE3) cells (Invitrogen) by inducing at $OD_{600}$ = 0.4–0.6 with 1 mM IPTG for 3 hr at 37°C. Cells were resuspended in 15 mL lysis buffer (50 mM sodium phosphate monobasic (pH 8), 300 mM NaCl, 10 mM imidazole, 1 mM PMSF, 1 mM DTT) per 600 mL culture and lysed by sonication. Cell debris was removed by centrifugation, loaded onto a 5 mL HisTrap HP Ni-NTA column (GE healthcare), and eluted with elution buffer (50 mM sodium phosphate monobasic (pH 8), 300 mM NaCl, 250 mM imidazole, 1 mM PMSF, 1 mM DTT). Peak fractions were concentrated and further purified

with a superdex s200 10/300 size exclusion column that was equilibrated with the Reb1 storage buffer (20 mM HEPES-NaOH (pH 7.5), 350 mM NaCl, 1% Tween-20). Pure fractions, as determined by coomassie SDS PAGE, were pooled and concentrated, before glycerol was added to a final concentration of 10%. The samples were flash-frozen and stored at −80°C.

## Preparation of Cbf1

Cbf1 was a gift from S Diekmann and was expressed and purified as previously described (*Kuras et al., 1997*; *Wieland et al., 2001*). Briefly, Cbf1 was cloned into pET28a and expressed in *Escherichia coli* BL21(DE3) cells (Invitrogen) by inducing at $OD_{600\ nm}$ = 0.4–0.6 with 0.5 mM IPTG for 4 hr at 37°C. Cells were resuspended in buffer A (50 mM $Na_2HP0_4$ (pH 7.5), 300 mM NaCl, 5 mM imidazole, 10% glycerol, 1 mM PMSF, 20 ug/mL pepstatin, 20 ug/mL leupeptin), lysed by sonication, and cell debris was removed by centrifugation (4 C, 23,000 x G, 20 min). After centrifugation, lysate was loaded onto a 5 mL HisTrap HP Ni-NTA column (GE Healthcare) and washed with 40 mL buffer A, 120 mL buffer B (50 mM $Na_2HP0_4$ (pH 7.5), 300 mM NaCl, 60 mM imidazole, 10% glycerol, 1 mM PMSF, 20 ug/mL pepstatin, 20 ug/mL leupeptin), and eluted with with buffer C (50 mM $Na_2HP0_4$ (pH 7.5), 300 mM NaCl, 340 mM imidazole, 10% glycerol, 1 mM PMSF, 20 ug/mL pepstatin, 20 ug/mL leupeptin). Pure fractions (as determined by SDS PAGE) were pooled, and imidazole was removed by washing with Buffer D (50 mM $Na_2HP0_4$ (pH 7.5), 300 mM NaCl, 10% glycerol, 1 mM PMSF, 20 ug/mL pepstatin, 20 ug/mL leupeptin) in a 10 K amicon (Millipore).

## Preparation of DNA molecules

DNA molecules for PIFE, FRET, and EMSA experiments were prepared by PCR with Cy3/Cy5/biotin-labeled oligonucleotides (Sigma) from a plasmid containing the 601 nucleosome positioning sequence (NPS) with a consensus Reb1- or Cbf1-binding site at various positions. For Reb1 experiments, the potential Reb1-binding site at positions 87–93 of the 601 was removed by site-directed mutagenesis. Oligonucleotides (*Supplementary file 1* Table S4)) were labeled with Cy3 or Cy5 NHS ester (GE Healthcare) at an amino group attached at the 5'-end or at an amine-modified dT, and purified by HPLC with a 218TP C18 column (Grace/vydac). Following PCR amplification, DNA molecules were purified using a MonoQ column (GE Healthcare).

## Preparation of histone octamers

Human recombinant histones were expressed and purified as previously described (*Luger et al., 1999*). Expression vectors were generous gifts from Dr. Karolin Luger (University of Colorado) and Dr. Jonathan Widom. Mutation H3(C110A) was introduced by site-directed mutagenesis (Agilent). The histone octamer was refolded by adding each of the histone together at equal molar ratio and purifying as previously described (*Luger et al., 1999*). H2A(K119C)- and H3(V35C)-containing histone octamer were labeled with Cy5-maleamide (GE Healthcare) as previously described (*Shimko et al., 2011*).

## Preparation of nucleosomes

Nucleosomes were reconstituted from Cy3-labeled DNA and purified Cy5-labeled histone octamer by double salt dialysis as previously described (*Shimko et al., 2011*). Dialyzed nucleosomes were loaded onto 5–30% sucrose gradients and purified by centrifugation on an Optima L-90 K Ultracentrifuge (Beckman Coulter) with a SW-41 rotor. Sucrose fractions containing nucleosomes were collected, concentrated, and stored in 5x TE (pH 8) on ice.

## Electrophoretic mobility shift assays

0.5 nM DNA or nucleosomes were incubated with 0–100 nM Reb1 in 10 mM Tris-HCl (pH 8), 130 mM NaCl, 10% glycerol, 0.0075% v/v Tween-20 for at least 5 min and then resolved by electrophoretic mobility shift assay (EMSA) with a 5% native polyacrylamide gel in 3x TBE.

## Ensemble PIFE measurements

Reb1 binding to its target site on Cy3-DNA was determined by protein-induced fluorescence enhancement (PIFE) (*Hwang et al., 2011*), in which Cy3 fluorescence increases upon protein binding. Fluorescence spectra were acquired with a Fluoromax4 fluorometer (Horiba) using an excitation

wavelength of 510 nm. 0.5 nM DNA was incubated for at least 5 min with 0–300 nM Reb1 in 10 mM Tris-HCl (pH 8), 130 mM NaCl, 10% glycerol, and 0.0075% v/v Tween-20. Fluorescence spectra were analyzed using Matlab to determine the change in Cy3 fluorescence.

## Ensemble FRET measurements

Reb1 binding to Cy3-Cy5 nucleosomes was measured as previously described (*Li and Widom, 2004*; *Shimko et al., 2011*). 0.5 nM nucleosomes were incubated for at least 5 min with 0–300 nM Reb1 in 10 mM Tris-HCl (pH 8), 130 mM NaCl, 10% glycerol, and 0.0075% v/v Tween-20. Fluorescence emission spectra were acquired as previously described (*Shimko et al., 2011*). FRET efficiency was measured using the (Ratio)$_A$ method (*Clegg, 1992*). Using EMSAs of Reb1 binding to P8 nucleosomes that contain unlabeled histone octamer (*Figure 1—figure supplement 6*), we confirmed that the Cy5 fluorophore causes a $1.2 \pm 0.1$ lower $S_{1/2}$ with P8 nucleosomes and therefore does not impact Reb1 binding to nucleosomes.

## Single-molecule TIRF microscopy

The smTIRF microscope was built on an inverted IX73-inverted microscope (Olympus) as previously described (*Roy et al., 2008*). 532 nm and 638 nm diode lasers (Crystal Lasers) were used for Cy3 and Cy5 excitation. The excitation beams were expanded and then focused through a quartz prism (Melles Griot) at the surface of the quartz flow cell. A 1.3 N.A. silicone immersion objective (Olympus) was used to collect fluorescence, which was separately imaged onto an iXon3 EMCCD camera (Andor) with a custom-built emission path containing bandpass filters and dichroic beam splitters (Chroma Tech). Each video was acquired using Micro-Manager software (Open Imaging) (*Edelstein et al., 2014*).

## Flow cell preparation

Flow cells were functionalized as previously described (*Kinz-Thompson et al., 2013*). Briefly, quartz microscope slides (Alfa Aesar) were sonicated in toluene and then ethanol, and then further cleaned by piranha (3:1 mixture of concentrated sulfuric acid to 50% hydrogen peroxide). Slides were washed in water and, once completely dry, incubated in 100 uM mPEG-Si and biotin-PEG-Si (Laysan Bio) overnight in anhydrous toluene. Functionalized quartz slides and coverslips were assembled into microscope flow cells using parafilm with cut channels. Before each experiment, the flow cell was treated sequentially with 1 mg/ml BSA, 40 ug/ml streptavidin, and biotin-labeled DNA or nucleosomes.

## Single-molecule fluorescence measurements of Reb1- or Cbf1-binding kinetics

Biotinylated sample molecules (DNA or nucleosomes) were allowed to incubate in the flow cell at room temperature for 5 min and then washed out with imaging buffer containing the desired concentration of Reb1. The samples were first exposed to 638 nm excitation to determine the location of Cy5-labeled molecules and then to 532 nm excitation for both FRET and PIFE measurements. The imaging buffer for FRET experiments contained 10 mM Tris-HCl (pH 8), 130 mM NaCl, 10% glycerol, 0.5% v/v Tween-20, 0.1 mg/ml BSA, 2 mM Trolox, 0.0115% v/v COT, 0.012% v/v NBA, 450 ug/ml glucose oxidase (Sigma G2133) and 22 ug/ml catalase (Sigma C3155), while the imaging buffer for PIFE experiments contained 10 mM Tris-HCl (pH 8), 130 mM NaCl, 10% glycerol, 0.5% v/v Tween-20, 0.1 mg/ml BSA, 1% v/v BME, 450 ug/ml glucose oxidase (Sigma G2133) and 22 ug/ml catalase.

Single-molecule time series were fit to a two-state step function by the Hidden Markov Method using vbFRET (*Bronson et al., 2009*). Idealized time series were further analyzed using custom-written Matlab programs (*Source code 1*) to determine the dwell-time distributions of the TF bound and unbound states. 40% of traces were used in the analysis of FRET data and 13% of traces were used when analyzing PIFE data (*Supplementary file 1* Table S5). Dwell-time and unbound-time cumulative sum distributions were generated from these traces and each distribution was analyzed using MEMLET to determine the best fit for the data and ultimately to obtain rate constants for the transitions between bound and unbound states (*Woody et al., 2016*).

## FRAP assay

The yeast strains used for the FRAP experiment are diploids constructed by mating the haploid strains in BY4741 (MAT<u>a</u>) and BY4742 (MATα) backgrounds. GFP tags were introduced into BY4741 and BY4742 by integrating the *GFP–HIS3MX* cassette into the C-terminus of the target genes in the native genome (*Longtine et al., 1998*; *Huh et al., 2003*). See 'Key resources table' for strain information and *Supplementary file 1* table S6 for the primer list. Yeast cells containing GFP-labeled factors were cultured to log phase in synthetic medium, and then transferred onto an agar pad and mounted by a coverslip. Cells were imaged using the 60X lens of a FV1000 confocal microscope (Zeiss) at room temperature. A 488 nm laser was used to excite and bleach green fluorescence. Depending on the GFP intensity, 3–10% laser power was used to take the images; 100% power was used to bleach the samples. The photobleaching time was set to 0.2–0.5 s, and the intervals between consecutive frames were 0.25–5 s. Two frames were acquired before photobleaching, followed by 28–48 frames afterwards. The bleached region covered ~10–25% of the nuclear region. Images were analyzed using Fiji-ImageJ. The average fluorescence intensities of the bleached and unbleached regions were recorded, and the ratio between them was used in the recovery curve.

## Acknowledgements

We thank the members of the Poirier and Bai Labs for helpful discussions. This work was supported by NIH grants R01 GM121858 (to LB and MGP), R01 GM121966 (to MGP), and T32 GM086252 (to BTD) and by NSF grant 1516979 (to MGP).

## Additional information

### Funding

| Funder | Grant reference number | Author |
|---|---|---|
| National Institutes of Health | R01 GM121858 | Lu Bai<br>Michael G Poirier |
| National Institutes of Health | R01 GM121966 | Michael G Poirier |
| National Institutes of Health | T32 GM086252 | Benjamin T Donovan |
| National Science Foundation | 1516979 | Michael G Poirier |

The funders had no role in study design, data collection and interpretation, or the decision to submit the work for publication.

### Author contributions

Benjamin T Donovan, Conceptualization, Formal analysis, Investigation, Visualization, Writing—original draft, Writing—review and editing; Hengye Chen, Data curation, Formal analysis, Investigation, Writing—review and editing; Caroline Jipa, Data curation, Formal analysis, Investigation; Lu Bai, Conceptualization, Supervision, Funding acquisition, Project administration, Writing—review and editing; Michael G Poirier, Conceptualization, Supervision, Funding acquisition, Writing—original draft, Project administration, Writing—review and editing

### Author ORCIDs

Benjamin T Donovan http://orcid.org/0000-0003-0177-799X
Lu Bai http://orcid.org/0000-0003-3667-2944
Michael G Poirier http://orcid.org/0000-0002-1563-5792

### Decision letter and Author response

Decision letter https://doi.org/10.7554/eLife.43008.030
Author response https://doi.org/10.7554/eLife.43008.031

## Additional files

### Supplementary files

• Supplementary file 1. Tables S1 to S6 reporting all binding-affinity measurements, all measured rates, a comparison of the relative binding affinities (Nuc/DNA) in the ensemble and single molecule experiments, the primers used in in vitro experiments, the single-molecule quality control data, and the primers used in the in vivo experiments.
DOI: https://doi.org/10.7554/eLife.43008.024

• Source code 1. Matlab script for data analysis of idealized traces.
DOI: https://doi.org/10.7554/eLife.43008.025

• Transparent reporting form
DOI: https://doi.org/10.7554/eLife.43008.026

### Data availability

All analyzed data generated is included in the manuscript. In supplementary file 1, we include 6 supplementary tables. Table S1 documents all binding affinity measurements from this study. Table S2 documents measured binding rates from single molecule experiments. Table S3 documents relative binding affinities (Nuc/DNA) for ensemble and single molecule experiments. Table S4 documents the primers for in vitro experiments. Table S5 documents quality control information from single molecule experiments. Table S6 documents the primers used for FRAP experiments. Videos supporting this study have been deposited to Zenodo and are available under the doi:10.5281/zenodo.2595208.

The following dataset was generated:

| Author(s) | Year | Dataset title | Dataset URL | Database and Identifier |
|---|---|---|---|---|
| Donovan B | 2019 | Dissociation rate compensation mechanism for budding yeast pioneer transcription factors | https://doi.org/10.5281/zenodo.2595208 | Zenodo, 10.5281/zenodo.2595208 |

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
