## [Decision Letter]

Thank you for submitting your article "Dissociation rate compensation mechanism for budding yeast pioneer transcription factors" for consideration by *eLife*. Your article has been reviewed by three peer reviewers, including Tim Formosa as the Reviewing Editor and Reviewer #1, and the evaluation has been overseen by Jessica Tyler as the Senior Editor. The following individuals involved in review of your submission have agreed to reveal their identity: Maria Spies (Reviewer #3).

The reviewers have discussed the reviews with one another and the Reviewing Editor has drafted this decision to help you prepare a revised submission.

Summary:

Donovan et al. present a rigorous dissection of the properties of two pioneer DNA-binding factors from yeast, Reb1 and Cbf1, comparing their ability to bind their target sites when they are in free DNA to when they are in nucleosomes. The key findings are that (1) the affinity of the factors for sites in naked DNA and sites close to the entry-exit point of nucleosomes is comparable, even though the latter are less accessible; (2) these factors use mechanisms distinct from what has been described previously as binding is coupled to unpeeling of nucleosomal DNA without loss of histones or repositioning of the octamers; and (3) dissociation from the nucleosomal context is slowed, compensating for the decreased binding rate that results from limited accessibility and thereby maintaining the same affinity. The experiments are a well-designed and carefully executed mixture of ensemble and single-molecule approaches whose quantitative depth substantially surpasses previous studies of pioneer factors.

The reviewers all found the "off-rate compensation" model in the absence of changes to nucleosome structure or positioning to be well-supported and an important conceptual advance. The primary weakness noted was that a testable model for how the compensation works mechanistically was suggested but not examined further. As either outcome from this test would be informative, including it would enhance the impact of the manuscript significantly, bringing it into the range expected for publication in this journal.

Essential revisions:

New experiments:

1) The results presented suggest a model for which domain of Reb1 is responsible for adjusting the off-rate in different contexts, but the manuscript does not take the opportunity to test this model. Specifically, the authors imply that the negatively charged N-terminal domain of Reb1 may contact the exposed positively charged histone residues (subsection “Cbf1 also binds and dissociates from nucleosomes significantly slower than DNA” and Discussion section). This plausible and testable model could be addressed by measuring the binding and dissociation rate constants after deleting the N-terminal domain or mutating it to a neutral form. Any outcome is informative as it could either provide a mechanism for the altered dissociation rate and the stabilization of octamer composition and positioning or demonstrate that these are properties of the DNA-binding domain itself.

Changes to the manuscript:

2) The data on the second panel of Figure 3 (labeled D, should be E?) are fit to a double exponential, but only one phase is used to calculate the rate. While it is easy to envision why a double exponential fit was used for the ON time distribution (i.e. two distinct complexes with two stabilities), it is much harder to explain the second exponential for the OFF dwell times. The authors state that this phase is [Reb1]-independent and does not represent binding, but should speculate about the nature of this phase.

3) The affinity for a nucleosome without the recognition sequence only decreased about 10-fold. This non-specific, but quite significant, interaction with nucleosomes independent of the presence of the target DNA binding site needs to be discussed and incorporated into the overall model.

4) The relative changes in rate constants are the primary basis for the arguments presented, but because in most cases more and less prominent classes of events were observed, it can be confusing for the reader to understand which constants are being compared. For example, in subsection “Reb1 binds and dissociates from nucleosomes significantly slower than DNA” the authors state that "…the Reb1 dissociation rate from nucleosomes is also about 50-fold lower than from DNA." The two rates being compared seem to be 0.58 and 0.0044, which is a 132-fold reduction. The authors may be saying that 132-fold and 50-fold are similar, which is fine, but trying to determine whether this is the comparison they were making made the argument harder to follow. They could simply say that the rate decreased ~130-fold in this case, which is similar to the 53-fold change in the on-rate (as is done for Cbf1 in subsection “Cbf1 also binds and dissociates from nucleosomes significantly slower than DNA”). The actual comparisons could be added as a column to the table in Figure 7A; the general idea is to give the actual ratio explicitly then let the reader decide how similar the results are instead of making the reader wonder precisely what is being compared. The same argument should be applied to the dissociation constants calculated from the rate constants (subsection “Reb1 binds and dissociates from nucleosomes significantly slower than DNA”) and comparisons with EMSA (subsection “Cbf1 also binds and dissociates from nucleosomes significantly slower than DNA”).

5) The Materials and methods section does not describe how the yeast strains used were constructed. This is particularly important for understanding the FRAP results in which a large GFP construct was added to small histone and Nhp6A proteins. The details of this procedure (plasmid or integrated? N or C terminal fusions?) influence the interpretation and so should be given here.

---

## [Author Response]

Essential revisions:New experiments1) The results presented suggest a model for which domain of Reb1 is responsible for adjusting the off-rate in different contexts, but the manuscript does not take the opportunity to test this model. Specifically, the authors imply that the negatively charged N-terminal domain of Reb1 may contact the exposed positively charged histone residues (subsection “Cbf1 also binds and dissociates from nucleosomes significantly slower than DNA” and Discussion section). This plausible and testable model could be addressed by measuring the binding and dissociation rate constants after deleting the N-terminal domain or mutating it to a neutral form. Any outcome is informative as it could either provide a mechanism for the altered dissociation rate and the stabilization of octamer composition and positioning or demonstrate that these are properties of the DNA-binding domain itself.

As recommended by the reviewers, we prepared the Reb1 N-terminal deletion mutant, Reb1-ΔN, and investigated the impact of this truncation on ensemble binding and the rates of binding and dissociation. This directly tests the model that this highly acidic N-terminal domain is responsible for the pioneering property of Reb1 and for the 130-fold decrease in Reb dissociation rate from the nucleosome relative to DNA. First, using both EMSAs and ensemble fluorescence, we find that Reb1-ΔN binds its site within naked DNA with a 1.5-fold higher affinity than binding its target site within P8 nucleosomes. For comparison, wild type (WT) Reb1 binds its site within DNA with a 2.1-fold lower affinity than to P8 nucleosomes. This implies that even without the acidic N-terminal domain Reb1 continues to bind DNA and nucleosomes with similar affinities. This is in stark contrast to traditional transcription factors (TFs), which bind nucleosomes at the P8 position with orders of magnitude lower affinity. Therefore, we conclude that the acidic N-terminal domain of Reb1 is not required for the pioneering property of Reb1.

To then further investigate this acidic N-terminal domain, we determined with single molecule fluorescence measurements the kinetics of Reb1-ΔN binding to and dissociating from nucleosomes. We find that the Reb1-ΔN binding to nucleosomes is similar to WT Reb1 with a 2-fold slower rate, while Reb1-ΔN dissociates from nucleosomes with a nearly identical rate as full length Reb1. These single molecule rate measurements imply a 2-fold difference in the effective K_D_, while our ensemble FRET measurements observe a 5-fold difference in the S_1/2_. This 2.5-fold difference between the ensemble and single molecule measurements of Reb1-ΔN and WT Reb1 affinities to nucleosomes is within the typical range when we compare ensemble and single molecule data. More importantly, they conclusively show that the Reb1 acidic N-terminal domain is not responsible for the 130-fold decrease in dissociation rate from nucleosomes and confirms the ensemble studies that find Reb1-ΔN retains its pioneering property.

These experiments significantly enhance the manuscript by ruling out the possibility that the N-terminal tail is responsible for the pioneering mechanism of Reb1, which we suggested in the original submission. We are grateful to the reviewers for recommending this experiment. To incorporate these new results into the manuscript, we have added a new main figure (Figure 6), two new supplemental figures (Figure 6—figure supplement 1 and Figure 6—figure supplement 2), a new Results section and a paragraph to the Discussion section. We also removed the text in the Discussion section that suggests that the Reb1 N-terminal domain could be responsible for the 50-fold decrease in the dissociation rate from nucleosomes relative to DNA. Instead, we use the observation that the N-terminal domain is highly acidic to motivatie the N-terminal deletion experiment.

Changes to the manuscript:2) The data on the second panel of Figure 3 (labeled D, should be E?) are fit to a double exponential, but only one phase is used to calculate the rate. While it is easy to envision why a double exponential fit was used for the ON time distribution (i.e. two distinct complexes with two stabilities), it is much harder to explain the second exponential for the OFF dwell times. The authors state that this phase is [Reb1]-independent and does not represent binding, but should speculate about the nature of this phase.

Indeed, the smPIFE measurements of Reb1-DNA interactions determined that the primary low to high PIFE transition rate is concentration dependent, which we attribute to Reb1 binding to DNA of 0.0032 s-1nM^-1^. The additional minor low to high PIFE transition rate is independent of the Reb1 concentrations, which rules out the possibility that this transition is due to Reb1-DNA binding. Instead, it is likely a structural transition of the Reb1-DNA complex that results in a transition from a low to high PIFE state. PIFE involves interactions on the nm length scale (Hwang, et al., 2011), so structural changes on this length scale would be sufficient to cause a change in PIFE. This also implies that one of the transitions from the high to low PIFE states is not due to Reb1-DNA dissociation, but a structural transition as well. Since the major high to low PIFE transition (0.58 s^-1^) is consistent with the dissociation constant measured by ensemble PIFE measurements, we conclude that the minor high to low PIFE transition rate (0.036 s^-1^) is due to Reb1-DNA structural transitions. This interpretation is similar to previous studies of intrinsically disordered proteins, where one ON rate is concentration dependent and is interpreted as binding, while a second ON rate is concentration independent and interpreted as a structural transition (Dogan et al., 2012).

To include these data interpretations in the manuscript, we added more detailed description of the smPIFE transition rates in the Results section that reports the Reb1 smPIFE measurements. We also altered the description of the smPIFE data so that we are careful about how we refer to the PIFE transition rates. Only after comparing the smPIFE data to the ensemble PIFE data do we refer to the concentration dependent low to high transition rate as a binding rate. We also corrected the panel labeling in Figure 3.

3) The affinity for a nucleosome without the recognition sequence only decreased about 10-fold. This non-specific, but quite significant, interaction with nucleosomes independent of the presence of the target DNA binding site needs to be discussed and incorporated into the overall model.

The 10-fold difference between Reb1 specific and non-specific equilibrium binding is less than traditional transcription factors such as bacterial TF LacI (Lin and Riggs, 1975) and the eukaryotic TF Gal4 (Liang, et al., 1996), which bind their consensus sequence with orders of magnitude higher specificity than a non-target DNA site. In contrast, the most extensively characterized pioneer factor, FoxA, has only a 2-fold difference (Sekiay, et al., 2009). Therefore, the specificity of Reb1 is not out of line with other pioneer factors.

In addition, comparing TF binding to a nucleosome with and without its target sequence is complicated by the fact that there are potentially many non-specific binding sites. We do not have direct experimental data on the nature of the non-specific interactions of Reb1 with nucleosomes. However, non-specific binding to DNA occurs at a concentration of ~30 nM (Figure 1B and Figure 3C), which is similar to Reb1 non-specifically binding to nucleosomes. This suggests that Reb1’s non-specific interactions with nucleosomes is largely through interactions with the nucleosomal DNA that is facing away from the histone octamer surface. If the non-specific Reb1-nucleosome interactions is indeed with nucleosomal DNA, the dissociation rate is likely faster than from its binding site within DNA, which is already 130-fold faster than from its binding site within P8 nucleosomes. Therefore, the Reb1 dissociation rate when non-specifically bound to nucleosomes is expected to be more than 130-fold greater than when specifically bound to P8 nucleosomes.

The detailed mechanism of how Reb1 functions in vivo with a relatively low specificity is beyond the scope of the current paper. However, we have considered 2 mechanisms that fit with the dissociation rate compensation mechanism. The first mechanism is that Reb1 is kinetically regulated, where the residence time of the bound state instead of the probability Reb1 is bound is key for Reb1 function. Here, the idea is that the long residence time of Reb1 at its site within the edge of the nucleosomes is required for Reb1 to recruit transcription co-activators such as chromatin remodelers and histone modifying complexes to function, while the over 130-fold shorter residence time of Reb1 nonspecifically bound to nucleosome is too short of Reb1 to recruit and maintain the co-activators occupancy to perform its function. The second mechanism is that Reb1’s specificity could be enhanced by the binding of the complexes Reb1 recruits when Reb1 is bound to its site near the edge of the nucleosome. For example, this could happen for complexes that interact with the histone H3 N-terminal, which is located at where the DNA exits the nucleosome.

We have incorporated this discussion of Reb1 specificity to the Discussion section of the manuscript.

4) The relative changes in rate constants are the primary basis for the arguments presented, but because in most cases more and less prominent classes of events were observed, it can be confusing for the reader to understand which constants are being compared. For example, in subsection “Reb1 binds and dissociates from nucleosomes significantly slower than DNA” the authors state that "…the Reb1 dissociation rate from nucleosomes is also about 50-fold lower than from DNA." The two rates being compared seem to be 0.58 and 0.0044, which is a 132-fold reduction. The authors may be saying that 132-fold and 50-fold are similar, which is fine, but trying to determine whether this is the comparison they were making made the argument harder to follow. They could simply say that the rate decreased ~130-fold in this case, which is similar to the 53-fold change in the on-rate (as is done for Cbf1 in subsection “Cbf1 also binds and dissociates from nucleosomes significantly slower than DNA”). The actual comparisons could be added as a column to the table in Figure 7A; the general idea is to give the actual ratio explicitly then let the reader decide how similar the results are instead of making the reader wonder precisely what is being compared. The same argument should be applied to the dissociation constants calculated from the rate constants (subsection “Reb1 binds and dissociates from nucleosomes significantly slower than DNA”) and comparisons with EMSA (subsection “Cbf1 also binds and dissociates from nucleosomes significantly slower than DNA”).

As recommended, we have clarified all of the comparisons made in the manuscript so that we explicitly state what measurements are being compared and updated the table that is now in Figure 8.

5) The Materials and methods section does not describe how the yeast strains used were constructed. This is particularly important for understanding the FRAP results in which a large GFP construct was added to small histone and Nhp6A proteins. The details of this procedure (plasmid or integrated? N or C terminal fusions?) influence the interpretation and so should be given here.

We have added an additional Materials and methods section and a supplementary table explaining how the strains were constructed (See key resources table for strain information and Supplementary file 6 for primer list).